



# Redistribution of total reactive nitrogen in the lowermost Arctic stratosphere during the cold winter 2015/2016

Helmut Ziereis[1], Peter Hoor[2], Jens–Uwe Grooß[3], Andreas Zahn[4], Greta Stratmann[1,6], Paul Stock[1], Michael Lichtenstern[1], Jens Krause[2,7], Armin Afchine[3], Christian Rolf[3], Wolfgang Woiwode[4], Marleen Braun[4], Jörn Ungermann[3], Andreas Marsing[1,2], Christiane Voigt[1,2], Andreas Engel[5], Björn–Martin Sinnhuber[4], and Hermann Oelhaf[4]

[1] Institut für Physik der Atmosphäre, Deutsches Zentrum für Luft- und Raumfahrt, Oberpfaffenhofen

[2] Institut für Physik der Atmosphäre, Johannes-Gutenberg-Universität Mainz, Mainz, Germany

[3] Institut für Energie- und Klimaforschung – Stratosphäre (IEK-7), Forschungszentrum Jülich, Jülich, Germany

[4] Institut für Meteorologie und Klimaforschung, Karlsruher Institut für Technologie, Karlsruhe, Germany

[5] Institut für Atmosphäre und Umwelt, Goethe Universität Frankfurt, Frankfurt, Germany

[6] now at Deutsches Elektronen–Synchrotron (DESY), Hamburg, Germany

[7] now at Excelitas Technologies GmbH & Co. KG, Wiesbaden, Germany

*Correspondence to*: Helmut A. Ziereis (helmut.ziereis@dlr.de)

**Abstract**. During winter 2015/2016 the Arctic stratosphere was characterized by extraordinarily low temperatures in connection with the occurrence of extensive polar stratospheric clouds. From mid of December 2015 until mid of March 2016 the German research aircraft HALO (High Altitude and Long–Range Research Aircraft) was deployed to probe the lowermost stratosphere in the Arctic region within the POLSTRACC (Polar Stratosphere in a Changing Climate) mission. More than twenty flights have been conducted out of Kiruna/Sweden and Oberpfaffenhofen/Germany, covering the whole

winter period. Besides total reactive nitrogen ($NO_y$), observations of nitrous oxide, nitric acid, ozone and water were used for this study. Total reactive nitrogen and its partitioning between gas- and particle phase are key parameters for understanding processes controlling the ozone budget in the polar winter stratosphere. The redistribution of total reactive nitrogen was evaluated by using tracer–tracer correlations. In January air masses with extensive nitrification were encountered at altitudes between 12 and 15 km. The excess $NO_y$ amounted up to about 6 ppb. During several flights, along with gas–phase

nitrification, indications for extensive occurrence of nitric acid containing particles at flight altitude were found. These observations support the assumption of sedimentation and subsequent evaporation of nitric acid containing particles leading to redistribution of total reactive nitrogen. Remnants of nitrified air masses have been observed until mid of March. Between end of February and mid of March also de–nitrified air masses have been observed in connection with high potential temperatures. Using tracer–tracer correlations, missing total reactive nitrogen was estimated to amount up to 6 ppb. This

indicates the downward transport of air masses that have been denitrified during the earlier winter phase. Observations within POLSTRACC, at the bottom of the vortex, reflect heterogeneous processes from the overlying Arctic winter stratosphere. The comparison of the observations with CLaMS model simulations confirm and complete the picture arising from the present measurements. The simulations confirm, that the ensemble of all observations is representative for the vortex–wide vertical $NO_y$–redistribution.




## 1 Introduction

Since the mid–1980s, observations in the Antarctic and later in the Arctic region have revealed unprecedented ozone loss in the polar stratosphere with beginning of the spring season (e.g. Farman et al., 1985; Müller et al., 1996; Waibel et al., 1999; Sinnhuber et al., 2000). The discovery of the so–called ozone hole was the starting point of extensive measurement

campaigns with research aircraft, balloons, satellites, and ground based instruments to study the processes that lead to this ozone decrease. It turned out that gas–phase chemistry alone is not sufficient to explain these observations. Heterogeneous reactions on polar stratospheric clouds were identified as key processes for the reactions involved (e.g. Crutzen and Arnold, 1986; Solomon, 1999; Lowe and MacKenzie, 2008;). Particle surfaces serve as platform to convert inactive halogen compounds into halogen species that are suitable to destroy ozone in catalytic cycles. Depending on temperature,

composition and physical state, different types of polar stratospheric clouds can be distinguished: liquid supercooled droplets, binary or ternary solutions (SBS, STS), nitric acid hydrates (NAD, NAT) and water ice particles (e.g. Fahey et al., 2001; Hoyle et al., 2013; Khosrawi et al., 2017; Tritscher et al., 2021).

In this sense, the conversion of gas–phase nitric acid into particle phase is of decisive importance. It does not only prepare the surface for heterogeneous reactions, it also removes nitric acid as reaction partner for processes deactivating chlorine

compounds. Heterogeneous reactions also enable the de–noxification of the stratosphere, the conversion of $NO_x$ to nitric acid. This process also contributes to the inhibition of chlorine deactivation (e.g. Solomon, 1990; Waibel et al., 1999). Particle formation is followed by sedimentation leading to irreversible removal of nitric acid. The removal of nitrogen compounds from the stratosphere allows continuing ozone destruction that increases with increasing illumination of the polar vortex. Although, temperatures in the Arctic winter stratosphere are usually higher than in the Antarctic, PSCs regularly

occur as well (e.g. Pitts et al., 2018). During several aircraft campaigns nitrate containing PSC particles have been observed at altitudes between 15 and 21 km (Northway et al., 2002; Voigt et al., 2005).

Denitrified regions resulting from particle sedimentation were found predominantly at elevations above 15 to 16 km (e.g. Waibel et al., 1999; Fahey et al., 2001; Popp et al., 2001; Jin et al., 2006; Woiwode et al., 2014). Observations at lower altitudes are rare. Denitrified regions are associated with regions of elevated nitrogen concentrations caused by evaporation

of sedimenting particles (e.g. Fischer et al., 1997; Hintsa et al., 1998; Waibel et al., 1999). During the winter, the denitrified air masses in the polar vortex sink down. Therefore, heterogeneous processes at higher altitudes can lead to nitrification and with a time lag to a denitrification of the UTLS. Reactive nitrogen species are key parameters in reaction cycles controlling ozone concentration (e.g. Hegglin et al., 2006; Stratmann et al., 2016). A redistribution of nitrogen oxides therefore affects the chemistry at the UTLS where even small changes may have a significant impact on the radiative properties of the

atmosphere (Riese et al., 2012).

In recent decades, the Arctic was the target of several intensive missions like SOLVE/THESEO in 2000 (Newman et al., 2002) or RECONCILE in 2009/2010 (von Hobe et al., 2013) just to give a few examples. During several winter seasons, substantial ozone loss was observed, (e.g. Sinnhuber et al., 2000; Rex et al., 2006; Sinnhuber et al., 2011). These measurement campaigns were followed by the POLSTRACC (Polar stratosphere in a Changing Climate) mission in winter

2015/2016. POLSTRACC aimed to study the lower Arctic stratosphere over a full winter–spring period (Oelhaf et al., 2019). This winter was characterized by unusual low temperatures of the Arctic polar vortex. The analysis of different data sets showed that the early winter of this season was the coldest in the Arctic stratosphere in the last 68 years (Matthias et al., 2016). The temperatures in the lower stratosphere were close to or at record low values between late December and early February (Manney and Lawrence, 2016). In extended areas of the Arctic stratosphere, temperatures dropped below the

existence temperature of ice (Voigt et al., 2018). With the space–borne Lidar CALIOP on board of the CALIPSO satellite extended regions of polar stratospheric clouds have been observed between 15 and 26 km from December until end of January (Pitts et al., 2018; Voigt et al., 2018). Extensive PSC abundance in the Arctic lead to the activation and repartitioning of chlorine species (Johansson et al., 2018; Johansson et al., 2019; Marsing et al., 2019). Trace gas



measurements performed with the Aura Microwave Limb Sounder (MLS) showed remarkable denitrification in the polar

vortex (Manney and Lawrence, 2016). A finding that was also illustrated by model simulations. With the EMAC atmospheric chemistry–climate model a strong denitrification of 4 to 8 ppb was simulated (Khosrawi et al., 2017). Beginning of March, a major final warming began and the full breakdown of the vortex occurred at the beginning of April (Manney and Lawrence, 2016).

POLSTRACC has been an extensive measurement campaign using the German research aircraft HALO (High Altitude and

Long–Range Research Aircraft) as platform for in situ and remote sensing instruments. POLSTRACC was part of a combined mission called PGS (POLSTRACC–GW–LCYCLE–SALSA) including also objectives with respect to gravity waves and to stratosphere–troposphere exchange. One major goal was to study the interrelationship between climate and the polar stratosphere (Oelhaf et al., 2019).

The investigation of the reactive nitrogen distribution at the bottom of the polar vortex and the search for nitrate containing

particles was a key issue of this mission, which was pursued with in situ and remote sensing instruments (Braun et al., 2019). With the AENEAS (Atmospheric Nitrogen Oxides Measuring System) in situ instrument the total reactive nitrogen distribution was observed in the lowermost stratosphere. Observations of $N_2O$ and $O_3$ have been used to interpret the nitrogen oxides measurements. Tracer–tracer correlations are an important tool for studying processes apart from transport and mere gas–phase chemistry.

The observation period provided an unique opportunity to study the lowermost stratosphere over an entire winter period. So, the questions could be addressed: How does the distribution of reactive nitrogen evolve from late December to mid–March? What influence do heterogeneous processes have on the distribution of reactive nitrogen compounds at the bottom of the polar vortex? This extremely cold winter favoured the formation of PSC particles and their sedimentation. So far, PSC particles containing reactive nitrogen compounds have been found almost exclusively at altitudes above 15 km. Therefore,

another goal of these measurements was the search for PSC particles at flight altitude of HALO.

In addition, the observed reactive nitrogen is compared with CLaMS model simulations. It is investigated how the model–measurement comparison performs in the different phases of the winter and how well the model describes the different influences of heterogeneous processes on the nitrogen oxide distribution at the lowermost stratosphere.

## 2 Instruments and Methods

The observation period with HALO extended from end of December 2015 to mid–March 2016. The deployment of HALO was divided into three phases: Early, mid and late winter. During the first phase in December two flights were conducted out of Oberpfaffenhofen to probe the UTLS at mid and high latitudes. The mid–winter observation phase started on 12 January with a flight from Oberpfaffenhofen to Kiruna and ended with a flight out of Kiruna and back on 2 February. The observations were suspended until end of February. On 26 February, flights were resumed. The late–winter mission phase

ended with a flight on 18 March from the HALO home base in Oberpfaffenhofen to Kiruna and back.

During the three mission phases 21 science flights (some with intermediate landings) with more than 150 flight hours were performed. Most of the flights have been conducted out of Kiruna in Northern Sweden. More than 70 % of the data have been obtained north of about 60° N. Typical flight altitudes ranged between about 12.5 and 14.5 km in the lower stratosphere. A detailed description of the flights is given in Oelhaf et al. (2019).

### 2.1 Measurement Techniques


The HALO research aircraft (https://www.dlr.de/content/en/missions/halo.html) is based on a Gulfstream G550 large business aircraft with a maximum ceiling of about 15 km and range of more than 8000 km. The payload of nearly three tons





comprised a set of remote sensing and in situ instruments complementing each other. A detailed description can be found elsewhere (Oelhaf et al., 2019). For the present analysis experimental data from several instruments have been used.

### 2.1.1 Total reactive nitrogen (gas– and particle–phase)

Total reactive nitrogen ($NO_y$) is the sum of all reactive nitrogen species in the atmosphere namely NO, $NO_2$, $HNO_3$, PAN, $HNO_2$, $HNO_4$, $N_2O_5$, $ClONO_2$, and others. During POLSTRACC total reactive nitrogen was measured using the AENEAS (Atmospheric Nitrogen oxides measuring system) instrument. Since 2012 this measuring system has been regularly operated on HALO during several missions (Jurkat et al., 2016; Wendisch et al., 2016; Voigt et al., 2017; Lelieveld et al., 2018).

The detection of total reactive nitrogen is based on a well–established technique, comprising catalytic conversion and chemiluminescence. $NO_y$ species are reduced catalytically on the surface of a heated gold tube to NO e.g. (Bollinger et al., 1983; Fahey et al., 1985). As reducing agent hydrogen is added. Subsequently, NO is detected by a chemiluminescence detector e.g. (Ridley et al., 1974; Kley, 1980; Drummond et al., 1985). At DLR this detector type has also been used for observations from other aircraft like the DLR Falcon (Ziereis et al., 2000b) and the DLR Dornier 228 (Reiner et al., 1999). A modified instrument was used on the Russian research aircraft Geophysica (Voigt et al., 2005; Voigt et al., 2006; Molleker et al., 2014). For more than 15 years a NO/ $NO_y$ detector identical in construction has been operated on a commercial airliner in the framework of IAGOS–CARIBIC (In–service Aircraft for a Global Observing System: https://www.iagos.org/iagos–caribic/) (Stratmann et al., 2016).

During POLSTRACC, AENEAS was operated with two separate detector channels. Both channels of the instrument were equipped with gold converters allowing the detection of $NO_y$. The two separate $NO_y$–channels were connected to a forward– and aft–facing inlet, respectively. With the backward facing inlet mainly gas–phase total reactive nitrogen is measured. The sampling of particles larger than about 1 µm is discriminated (Feigl et al., 1999). The overall uncertainty of the total reactive nitrogen measurement depends on the actual ambient concentration. It is about 8 % for volume mixing ratios of 0.5 ppb and about 6.5 % for about 1 ppb (Stratmann et al., 2016; Oelhaf et al., 2019).

The forward–facing inlet oversamples particles. The oversampling is caused by the sub–isokinetic sampling of particles. Due to the high ratio between true air speed of the aircraft and the flow velocity inside the inlet line particles are sampled with enhanced efficiency relative to the gas–phase. This approach has been already used during earlier observations, for example from NASA ER–2 (Fahey et al., 2001; Northway et al., 2002), NASA DC–8 (Weinheimer et al., 1998) or the DLR Falcon (Feigl et al., 1999; Ziereis et al., 2004). It was used to investigate the nitrate content of PSC particles as well as that of cirrus ice particles. The enhancement factor depends on the flow ratio as well as on ambient pressure, temperature and particle size. The corresponding relation was derived by Belyaev and Levin (Belyaev and Levin, 1974) and was adapted for aircraft observations (Fahey et al., 1989; Feigl et al., 1999). The enhancement factor strongly increases with increasing particle diameter and yields a maximum value for diameters larger than about 10 µm. For larger particles the enhancement factor increases only slightly with diameter.

Whenever the expression "$NO_y$" is used in this study, gas–phase $NO_y$ detected with the backward facing inlet is meant. Total reactive nitrogen observed with the forward–facing inlet comprises gas–phase $NO_y$ and enhanced particle $NO_y$ ($NO_y\_P$) as gas–phase equivalent. It is denoted as $NO_y\_tot$. $NO_y\_tot$ is not corrected for oversampling. Apart from episodes when particles containing nitrogen oxides compounds where sampled with the forward–facing inlet, the $NO_y$ signal detected with the two channels agreed within about 7 %.

$$NO_y\_tot = EF^* NO_y\_P + NO_y \qquad (1)$$

EF=enhancement factor

$NO_y\_net$ is defined as the difference between total $NO_y$ and gas–phase $NO_y$. Particulate nitrate $NO_y\_P$ as gas–phase equivalent can be derived from the difference between the signal obtained with the forward ($NO_y\_tot$) and aft–facing inlet ($NO_y$) and the enhancement factor, respectively



$NO_y\_P = NO_y\_net/EF$     (2)

### 2.1.2 Nitrous oxide

The measurement of nitrous oxide ($N_2O$) is based on a quantum cascade laser infrared absorption spectrometer. The TRIHOP instrument was operated during POLSTRACC with a precision of 1.84 ppb and total uncertainty of 2.7 ppb for the measurement of $N_2O$ (Krause et al., 2018; Oelhaf et al., 2019).

### 2.1.3 Ozone

Ozone was measured by the Fast and Accurate In Situ Ozone Instrument (FAIRO). It combines UV–photometer and fast and precise chemiluminescence detector. The total uncertainty is 1.5 % and the typical precision is 0.5 % at 10 Hz (Zahn et al., 2012; Oelhaf et al., 2019).

### 2.1.4 Water vapour

Water vapour was measured by the Fast In Situ Stratospheric Hygrometer (FISH) based on a Lyman–alpha photometer fluorescence technique and achieved a precision and accuracy of 1 % and 5.6 % during POLSTRACC, respectively (Zöger et al., 1999; Oelhaf et al., 2019).

### 2.1.5 Nitric acid

A separate measurement of gas phase nitric acid ($HNO_3$) was performed by the Airborne chemical Ionization Mass

Spectrometer (AIMS) via another backward facing inlet. During POLSTRACC, the accuracy was 16 % with a precision of 10–15 % at a time resolution of 1.7 s (Jurkat et al., 2017; Marsing et al., 2019; Oelhaf et al., 2019).

### 2.1.6 GLORIA

Vertical distributions of nitric acid were retrieved from limb–imaging observations by the GLORIA instrument (Gimballed Limb Observer for Radiance Imaging of the Atmosphere; (Friedl–Vallon et al., 2014; Riese et al., 2014). Characteristics of

the GLORIA 2D chemistry mode and dynamics mode data presented here are discussed by (Johansson et al., 2018) and (Krasauskas et al., 2020). Note that GLORIA samples air volumes to the right–hand side of the aircraft and not along the vertical projection of the flight path. Furthermore, due to the limb viewing geometry, which enables detection of minor atmospheric constituents with low abundances, the individual GLORIA profiles constitute horizontally smoothed representations of the atmospheric scenery perpendicular to the flight path. Thus, the GLORIA data at flight altitude does not

exactly match the geolocations and sampling characteristics of the simultaneous in situ observations on board HALO. Therefore, in the presence of small–scale structures and horizontal trace gas gradients, differences between these different types of observations at flight altitudes are possible and are not necessarily indicative of instrument errors.

### 2.2 CLaMS

For the interpretation of the observations and the underlying processes, we employ simulations of the Chemical Lagrangian

Model of the Stratosphere (CLaMS). The chemical transport model CLaMS is based on the Lagrangian transport concept and is described elsewhere (Grooß et al., 2014 and references therein). The Lagrangian concept is used in two ways. First the chemical composition of the air is simulated for so–called air parcels that follow the wind and are distributed irregularly in space. Second, also the particles are simulated by the Lagrangian principle. The simulations follow multiple representative NAT particle parcels, which are transported by the wind and in addition are exposed to gravitational settling. As

temperatures increase above $T_{NAT}$, the sedimented particle parcels evaporate and cause nitrification of the surrounding air





parcels. Similarly, the vertical redistribution of water vapour by ice particles is included (Tritscher et al., 2019). This simulation setup has shown to reproduce the observed denitrification and nitrification (Grooß et al., 2014).

The CLaMS simulation is initialized on 1 November 2015 on the base of MLS satellite data, multi–annual CLaMS simulations (Pommrich et al., 2014) and tracer–tracer correlations using a similar procedure as described by Grooß et al. (2014) and runs until March 2016. The simulation encompasses the Northern Hemisphere from the surface to 900 K potential temperature with a vertical resolution of 100 m. Results of this simulation have also been shown elsewhere (Grooß et al., 2018; Braun et al., 2019; Johansson et al., 2019).

### 3 Observations

#### 3.1 Tracer–tracer correlations

Tracer–tracer correlations of long–lived species are an established method to study transformation processes of trace gases in the lower stratosphere. The relation between chemical species in the stratosphere is linear and compact as long as their chemical life time is long compared to transport time scales (Plumb and Ko, 1992). The relation between total reactive nitrogen and nitrous oxide can be used in this sense because their lifetime is long compared to transport time–scales (Keim et al., 1997).

In the present study the redistribution of total reactive nitrogen in the lowermost stratosphere during POLSTRACC was analysed using tracer–tracer correlations, namely the correlation between total reactive nitrogen and nitrous oxide. Photolysis and the reaction of nitrous oxide with O(1D) are the main sources of total reactive nitrogen in the stratosphere (Keim et al., 1997; Greenblatt and Ravishankara, 1990). These processes lead to the formation of nitric oxide. Subsequent reactions produce $NO_2$ and finally nitric acid ($HNO_3$) and other reactive nitrogen species. These processes are most effective in regions with high UV–radiation as in the tropical stratosphere (Murphy et al., 1993).

The correlation between total reactive nitrogen and nitrous oxide is conserved during the transport of air masses from the tropics to the Polar Regions as long as no sinks or sources for $NO_y$ are effective. The observed negative slope of the relation between $NO_y$ and $N_2O$ can be understood as a kind of conversion efficiency. It is the portion of $N_2O$ that is converted into total reactive nitrogen by photolysis and subsequent reactions and lies in the order of 6–8 % e.g. (Fahey et al., 1990b; Loewenstein et al., 1993; Weinheimer et al., 1993; Fischer et al., 1997; Strahan, 1999). This correlation is linear over a wide range of $N_2O$ concentrations. Above about 25 to 30 km altitude the loss reaction of NO with N becomes increasingly important leading to a deviation from the linear relationship for $N_2O$ values below about 100 ppb (Loewenstein et al., 1993). Observations within the lowermost stratosphere of the Northern Hemisphere suggest some seasonality of this slope (Hegglin et al., 2006). A stronger seasonality was observed in the tracer–tracer correlation between $O_3$ and $N_2O$ (Hegglin and Shepherd, 2007; Bönisch et al., 2011)

An analysis for a large number of ER–2 flights showed that the $NO_y$–$N_2O$ correlation is linear down to about 170 ppb (Strahan, 1999). Based on these observations a semi–empirical quantity called $NO_y*$ can be derived, estimating the concentration of expected $NO_y$ arising from observed $N_2O$ in the stratosphere.

Strahan (1999) formulated this relation–ship as follows.

$$NO_y* = (N_2O(ts) - N_2O(obs))*f + NO_y(ts) \qquad (3)$$

$N_2O(obs)$ is the observed $N_2O$ concentration, $N_2O(ts)$ is the tropospheric concentration of $N_2O$ entering the stratosphere and f is the conversion efficiency. An additional term was added accounting for the contribution of tropospheric $NO_y$ to the observed concentrations in the stratosphere.

As long as there are no additional processes in the lower stratosphere affecting the lifetime of $NO_y$ or $N_2O$, observed $NO_y$ should be very close to $NO_y*$ (within the uncertainty range of observations).

$dNO_y$ is defined as difference between calculated $NO_y*$ and observed gas–phase $NO_y$.





$$dNO_y = NO_y - NO_y^* \qquad (4)$$

In the lower winter polar stratosphere deviations of $dNO_y$ from zero can be indicative for processes like nitrification and denitrification, resulting from the formation of polar stratospheric cloud particles, their sedimentation and subsequent

evaporation. This relationship was derived from observations in stratospheric air masses; it is not expected to hold in tropospheric air masses.

### 3.2 $NO_y$–$N_2O$ observations during the POLSTRACC mission

### 3.2.1 Early winter phase

During the early winter phase of the POLSTRACC mission two flights have been performed out of Oberpfaffenhofen on 17

and 21 December 2015. The first flight headed from Oberpfaffenhofen to the west and north of Scotland. The flight on 21 December leaded to Spitzbergen with the main objective to perform a polar vortex survey in early winter. During this flight (at 12 to 14.4 km altitude) observed $NO_y$ concentrations ranged between about 1 and 3.4 ppb (Figure 1). The highest concentrations have been found at the northern turn–around point of the flight at about 81°N. As expected for undisturbed conditions, $NO_y$ and $N_2O$ are anticorrelated (Figure 1).

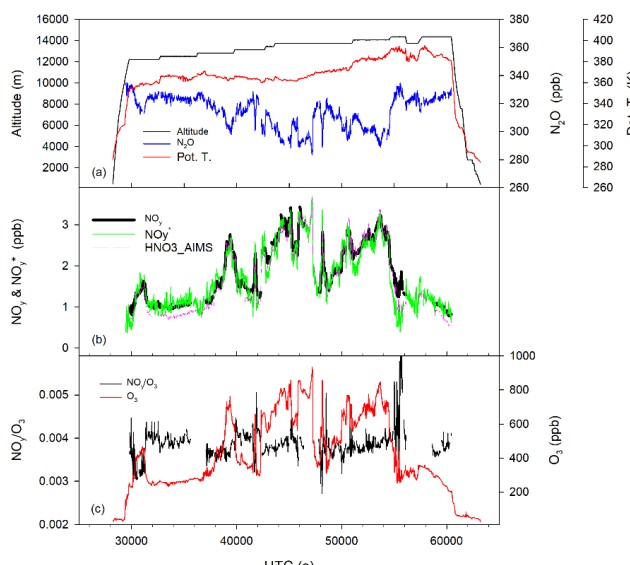


**Figure 1. PGS–flight 5 on 21 December 2015. (a) Altitude and $N_2O$, (b) $NO_y$ (observed) and $NO_y$*(calculated) and $HNO_3$ (AIMS–instrument), (c) $NO_y/O_3$ ratio and $O_3$.**

Similar $NO_y$ concentrations, with values between about 1 and 2 ppb, have also been found during earlier aircraft missions in

the winter Arctic lower stratosphere (Hubler et al., 1990; Weinheimer et al., 1993; Arnold et al., 1998). During the POLSTAR I mission in January 1997 $NO_y$ values up to 4 ppb have been observed (Ziereis et al., 2000a). In the winter polar stratosphere $NO_y$ is mainly comprised (>= 90 %) of nitric acid as was found by aircraft and balloon–borne observations (Wetzel et al., 2002; Schneider et al., 1999). For comparison, nitric acid observed with the AIMS mass spectrometer (Jurkat et al., 2016; Jurkat et al., 2017; Marsing et al., 2019) is also included in Figure 1. During most parts of the flight, nitric acid

and total reactive nitrogen agree with each other within the uncertainty range of both instruments, which confirms that stratospheric $NO_y$ is mainly dominated by $HNO_3$, while other $NO_y$ components may gain importance near the tropopause.





Figure 6a shows total reactive nitrogen plotted versus $N_2O$ for the flight on 21 December. To exclude tropospheric values that would affect the correlation, only values for nitrous oxide values lower than 320 ppb have been used for the analysis. A linear least squares fit gives a slope or conversion efficiency of $NO_y$ from $N_2O$ of about 0.064. This slope agrees reasonably

well with earlier observations performed with these instruments. In late summer 2012 the HALO mission TACTS (Transport and composition in the UT/LMS) (Müller et al., 2016) was performed at northern mid latitudes. Nitrification and denitrification could be excluded for this time of the year and region. A linear least squares fit between $NO_y$ and $N_2O$ for stratospheric values ($N_2O < 320$ ppb) gave a slope of about 0.067. The derived slope is also comparable to findings during earlier observations in the winter Arctic region that were not affected by nitrification or denitrification. During the AASE

missions in winter 1989 and 1991/1992 respectively, slopes between 0.064 and 0.078 have been observed (Fahey et al., 1990a; Fahey et al., 1990b; Weinheimer et al., 1993).

For further interpretation, the observations in the mid and late winter phase of POLSTRACC the slope or conversion efficiency of 0.067 (mid latitudes) was chosen as reference. The deviation from the value determined during the flight PGS– flight 5 can serve as a measure for the uncertainty in determining this slope, it is about 4 %. A further uncertainty in the

calculation of $NO_y*$ arises from the contribution of tropospheric $NO_y$. For the POLSTRACC observations $NO_y(ts)$ was estimated to be about 0.78 ppb (derived from the regression curve for tropospheric $N_2O$ values). This value lies well within the range spanned by previous observations. From the observations during the TACTS mission a tropospheric value of 0.65 ppb was derived. Strahan (1999) derived tropospheric $NO_y(ts)$ of 0.44+/-0.22 ppb by averaging measurements at the tropical tropopause. During STREAM–97 the observed mean $NO_y$ mixing ratios have been about 0.7 ppb in the upper troposphere

(Fischer et al., 2000). Observations with IAGOS–CARIBIC show a high variability of reactive nitrogen in the upper troposphere (Stratmann et al., 2016). Values depend on region and season where and when the observations were performed and range between about 0.4 and 1.4 ppb. The uncertainty in the estimation of $NO_y*$ resulting from the uncertainty of the tropospheric $NO_y$ contribution is highest directly at the tropopause and decreases with decreasing $N_2O$ concentration and increasing stratospheric character of the air mass. At $N_2O$ values of 300 ppb and less this uncertainty amounts to about 10 %

and less.

According to Eq. (3), the tropospheric concentration of $N_2O$ is included in the calculation of $NO_y*$. The tropospheric $N_2O$ value is steadily increasing over the years. Therefore, different air mass ages imply different tropospheric $N_2O$ concentrations. During POLSTRACC the age of the probed air masses ranged between about 1 and 5 years (Krause et al., 2018). Within five years the tropospheric $N_2O$ concentration increased by about 1.5 % (Combined Nitrous Oxide data from

the NOAA Global Monitoring Laboratory). This corresponds to a difference in tropospheric $N_2O$ of about 5 ppb and a difference in $NO_y*$ (assuming a conversion efficiency of 0.067) of about 0.3 ppb. Although this contribution to $NO_y*$ is comparatively small, the tropospheric $N_2O$ concentration in Eq. (3) was chosen according to its air mass age.

In Figure 1 measured $NO_y$ values are shown along with calculated $NO_y*$ values. During most of the time both curves agree well within the uncertainty range. A larger deviation at around 54000 s UTC was found at high $N_2O$ values close to the

tropopause where this relation is not expected to be valid.

Another diagnostic tool for characterizing the lowermost stratosphere with respect to reactive nitrogen species is the $NO_y/O_3$ ratio (Murphy et al., 1993). $NO_y$ and $O_3$ are mainly produced in the tropical stratosphere and have comparable lifetimes in the lower stratosphere. Their ratio is more constant than the concentration of the individual species itself. This can also be seen in Figure 1. $NO_y$ and $O_3$ exhibit a substantial variability along the flight track while the $NO_y/O_3$ ratio does not change a

lot. Typical values for this ratio in the lower stratosphere at high northern latitudes are around 0.003–0.004 for undisturbed conditions e.g. (Murphy et al., 1993; Fahey et al., 1996). Values of the same magnitude were observed during the POLSTRACC flight in December (Figure 1). Here a median $NO_y/O_3$ of about 0.0038 was measured. For comparison, in winter 1997 aircraft observations within the STREAM campaign in Arctic found values between 0.003 and 0.006 for this ratio in undisturbed stratospheric air (Fischer et al., 2000).

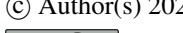



In December 2015, the polar vortex had already reached temperatures that allowed the formation of PSC particles (Oelhaf et al., 2019). With CALIOP large areas covered with PSC were observed between approximately 15 and 25 km altitude (Pitts et al., 2018). Based on the present study of the $NO_y$–$N_2O$ correlation and $NO_y/O_3$ ratio, no indications have been found that the lowermost Arctic stratosphere was already affected by redistribution of reactive nitrogen species at the beginning of the winter. However, considering that only one flight has been performed to the Arctic covering only a small part of the sub

vortex region, redistribution in the lowermost stratosphere already in December cannot be ruled out by these observations.

### 3.2.2 Mid–Winter Phase

The second phase of the POLSTRACC mission started with the transfer flight of HALO from Oberpfaffenhofen to Kiruna in northern Sweden on 12 January 2016. Seven flights from Kiruna were completed by 2 February 2016. More than 90 % of these flight routes were lying north of 60°N, with more than 85 % of the total flight time in the lower stratosphere with PV

values of more than 2 PVU.

During the midwinter phase the observed relation between $NO_y$ and $N_2O$ differs significantly from the situation in the early winter phase or from the observations at midlatitudes as during TACTS. This deviation was already observed during the first local flight out of Kiruna on 18 January. The flight went along the east side of Sweden and then along the coast of Norway back to Kiruna. The main scope of this mission flight was the probing of filamented stratospheric air (Oelhaf et al., 2019).

Significantly higher $NO_y$ concentrations have been observed at altitudes above 12 km than during the flight in December (see Figure 2). $NO_y$ values as high as 10 ppb were observed. In parallel, the $N_2O$ concentration was as low as 280 ppb. At the beginning and end of the flight $NO_y$ and $NO_y*$ were nearly identical. $NO_y$ and $NO_y*$ also agreed between about 41600 and 44000 UTC seconds when HALO flew close to the tropopause. Outside these periods observed $NO_y$ was significantly higher than calculated $NO_y*$. In Figure 6b $NO_y$ is shown versus $N_2O$ along with $NO_y*$. For stratospheric $N_2O$

concentrations, observed $NO_y$ levels were much higher than $NO_y*$, by up to about 6 ppb. This excess $NO_y$ cannot be explained by any unknown additional tropospheric source as the deviation from $NO_y*$ increases with decreasing $N_2O$. The excess $NO_y$ amounts up to about 50 % of the whole gas–phase total reactive nitrogen and is also reflected in the highly variable $NO_y/O_3$ ratio during this flight. High ratios have been found along with high values for $dNO_y$. Values changed from around 0.004 to values up to about 0.01. Observations with the GLORIA instrument on HALO complement the in situ

observations down to upper troposphere and also show nitrification of the lowermost stratosphere (Braun et al., 2019). Figure 2d shows a section through the atmosphere below HALO's flight altitude for the flight on 18 January. For this flight typical vertical resolution of ~500–1000 m was diagnosed for the chemistry mode data. A typical total uncertainties (1 σ) in the order of 10–20 % are estimated for this particular flight. Both, vertical resolution and total errors thereby depend on altitude and observed scenery.

In Figure 2b nitric acid obtained in the high spectra resolution "chemistry mode" of GLORIA for this flight is shown along with in situ observed total reactive nitrogen. Although airmasses probed with GLORIA and AENEAS are not identical, a high agreement between both measurements was found.





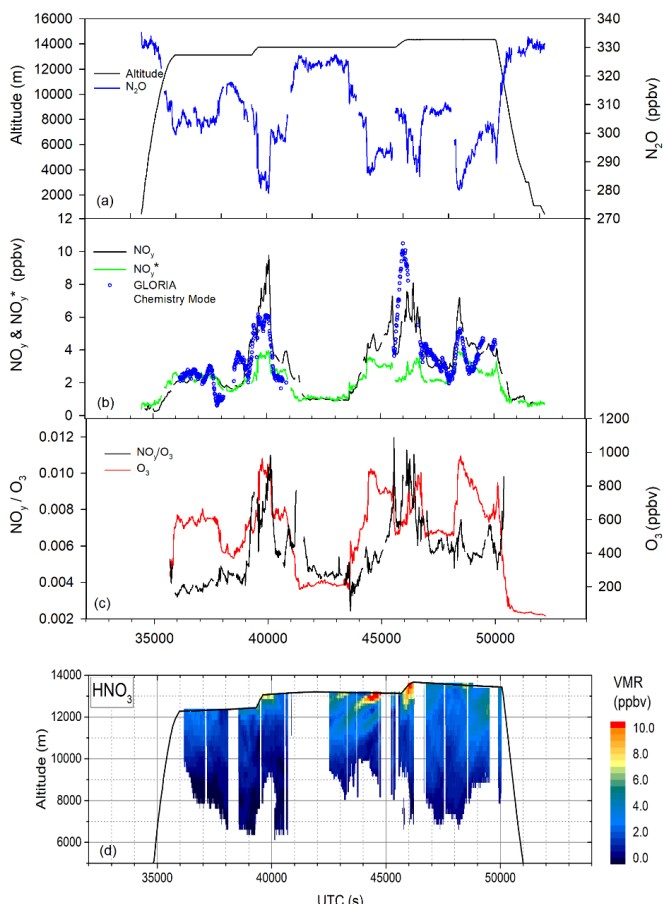

**Figure 2. PGS–flight 7 on 18 January, 2016. (a) Altitude and $N_2O$, (b) $NO_y$ (observed), $NO_y$*(calculated) and $HNO_3$ (GLORIA–**
**instrument), (c) $NO_y/O_3$ ratio and $O_3$, (d) $HNO_3$ observed with GLORIA.**

Substantial deviations of observed $NO_y$ from expected $NO_y$* have been observed during nearly all POLSTRACC flights of
the mid–winter phase. A further example for a flight with enhanced $NO_y$ is given in Figure 3. The flight on 31 January
leaded towards north of Scotland and Ireland and further north to about 76° N. West of Ireland and Scotland HALO dived
into the troposphere and encountered clean tropospheric airmasses with $NO_y$ and $O_3$ values down to about 0.2 and 30 ppb,
respectively. For this phase of flight, no $NO_y$* was determined because the above given relation is only valid for stratospheric
conditions. Between roughly 44000 and 52000 s UTC $N_2O$ values between about 300 and 260 ppb have been observed.
Concurrently, ozone and total reactive nitrogen increased (not shown). During this period observed $NO_y$ values have been up
to more than twice as high than estimated $NO_y$*. $dNO_y$ ranged between about two and more than 6 ppb. Again, higher $NO_y$
concentrations are also reflected in a higher $NO_y/O_3$ ratio. These findings suggest a substantial redistribution of total reactive
nitrogen the lower Arctic stratosphere. Particles containing total reactive nitrogen can sediment down to HALO flight
altitudes within a few days. The fall speed for particles with diameter of 10 µm and above is more than 1 km per day (Fahey
et al., 2001). Here, evaporation of the particles leads to increased $NO_y$ volume mixing ratios. The interpretation of this
observation as particle–based redistribution is supported by LIDAR observations from space and from HALO. CALIOP
detected PSCs in the Arctic stratosphere between December and late January at altitudes between 15 and 26 km (Pitts et al.,
360 2018).





Lidar observations onboard of HALO also show extensive regions of polar stratospheric clouds. On 22 January, a large ice PSC cloud was detected with a horizontal expanse of about 1400 km and thickness of up to 6 km between 18 and 24 km

altitude (Voigt et al., 2018). Observations of nitrification of the lower stratosphere up to more than 10 ppb by evaporating particles have also been made during previous airborne missions to the Arctic and Antarctic (Hubler et al., 1990; Fischer et al., 1997; Arnold et al., 1998; Dibb et al., 2006; Molleker et al., 2014).

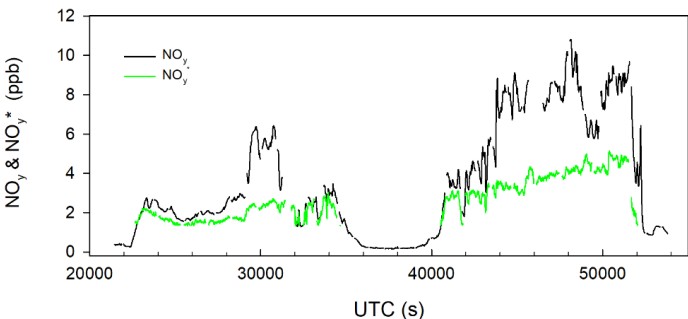

**Figure 3. PGS–flight 12 on 31 January, 2016. $NO_y$ (observed) and $NO_y^*$(calculated).**


The interpretation of the elevated level of $NO_y$ as remnants of evaporated particles during POLSTRACC is also supported by the observation of particulate nitrate at flight altitude. During several flights, indications for the occurrence of particles containing nitrate have been found (see Section 3.3).

**3.2.3 Late–Winter Phase**

The third mission phase covered the period between 26 February and 18 March 2016. In total 13 flights have been performed in late winter. Again, most of the flights leaded from Kiruna to regions north of 60°N. Late winter was characterized by descending air masses with lower $N_2O$ concentrations and higher potential temperature at HALO flight altitudes.

At the beginning of the third mission phase, the $NO_y$ distribution in lower stratosphere was characterized by the transition of

the influence of sedimentation and evaporation of particles to the influence by downward transported air masses that have been denitrified before. As an example, the flight on 26 February (Figure 4 and 6d) may serve. One objective of this flight was the observation in a stratospheric cold pool west of Greenland. On its way to Baffin Island total reactive nitrogen was generally higher than calculated $NO_y^*$. $dNO_y$ amounted up to about 4 ppb. Over Baffin Island $N_2O$ dropped down to values of about 205 ppb while concurrently potential temperature reached values of more than about 395 K (not shown). In these air

masses descended from the stratosphere above, observed $NO_y$ was about 4 ppb below calculated $NO_y^*$. On the way back to Kiruna, HALO again encountered air masses with observed $NO_y$ levels exceeding $NO_y^*$. Thus, during this flight, signatures of both nitrification and de–nitrification were observed, demonstrating the filamentous structure of the lower stratosphere.



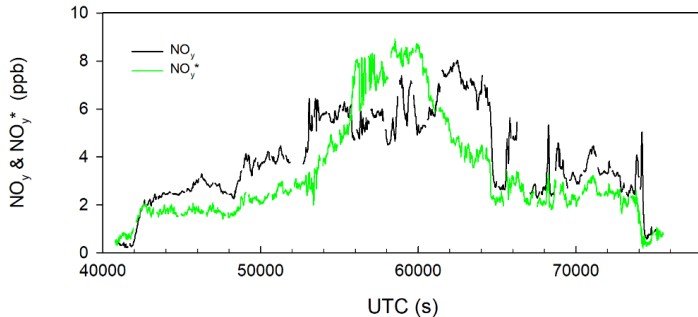

Figure 4. PGS–flight 14 on 26 February 2016. NO$_y$ (observed) and NOy*(calculated).


During the further course of the POLSTRACC mission the influence of the sedimentation and evaporation of particles on the composition of the lowermost stratosphere further decreased. On 13 March, the flight leaded from Kiruna to Kangerlussuaq in Greenland (Figure 5). Observed NO$_y$ was nearly identical with calculated NO$_y$* during large sections of the flight. During the last quarter of the mission at altitudes above 14 km air masses with potential temperatures between 380 and 410 K have

been encountered. Concurrently, N$_2$O dropped down to minimum values of about 180 ppb over the western part of Greenland. Calculated NO$_y$* exceeds observed NO$_y$ by up to 5ppb. During ascent and descend of HALO air masses close to the tropopause or below were encountered, where no correlation between NO$_y$ and N$_2$O is expected. Figure 6e shows NO$_y$ versus N$_2$O for this flight. Down to about 260 ppb N$_2$O, observed NO$_y$ and calculated NO$_y$* agreed within a reasonable uncertainty range. At lower N$_2$O concentrations, observed NO$_y$ values were significantly lower than calculated NO$_y$*. At

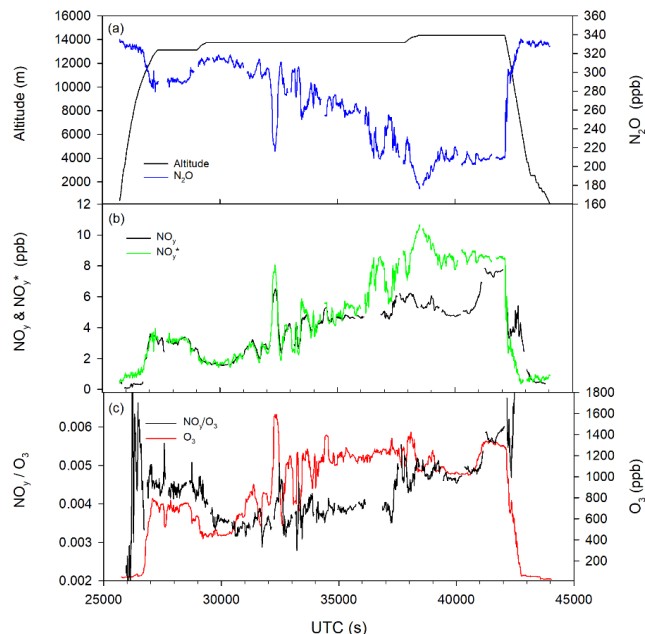


Figure 5. PGS–flight 19 on 13 March 2016. (a) Altitude and N$_2$O, (b) NO$_y$ (observed) and NO$_y$*(calculated), (c) NO$_y$/O$_3$ ratio and O$_3$.





around 180 ppb N$_2$O, about 50 % of the calculated NO$_y$* were missing. During the continuation of the flight from Kangerlussuaq to Oberpfaffenhofen in Germany (not shown), the potential temperature did not exceed 380 K and N$_2$O values remained above 260 ppb. During this flight no deviation of NO$_y$ from NO$_y$* was found. As mentioned earlier, besides the flight from Kangerlussuaq to Oberpfaffenhofen most of the flights stayed north of about 60°N. One exemption was the flight on 16 March to the Canary Islands. Within the accuracy of the measurement no significant deviation of observed NO$_y$

from NO$_y$* was found (not shown).  In situ observations from HALO revealing denitrified air masses fit well in the overall picture of this winter. They complete the picture arising from satellite observations and model simulations for this winter. Denitrified air masses have been observed with the Aura microwave limb sounder (MLS) (Manney and Lawrence, 2016). Simulations with the EMAC model showed both denitrified zones in the middle stratosphere and regions with enhanced total reactive nitrogen below for December to February. By mid of March the denitrified zone stretches down to pressures higher

than 100 hPa (Khosrawi et al., 2017).

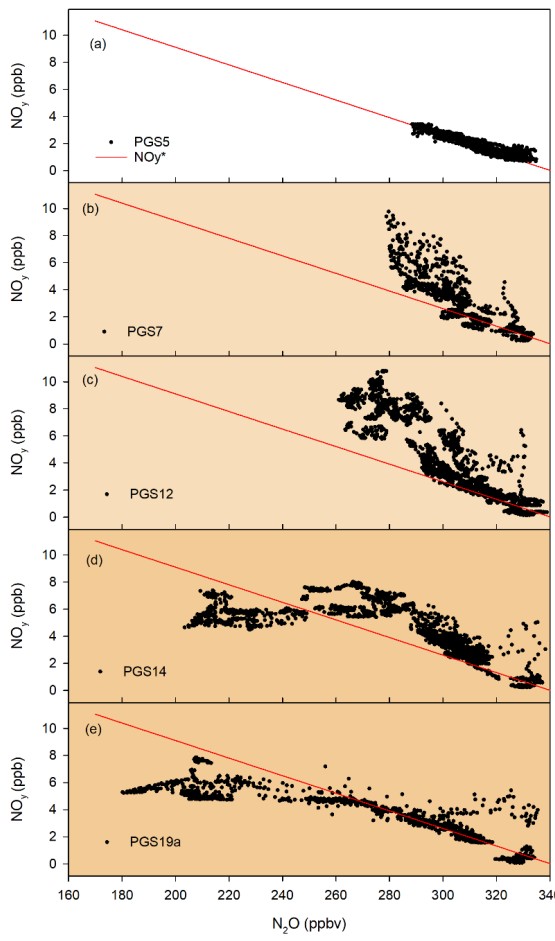

**Figure 6.** NO$_y$ and NO$_y$* versus N$_2$O for selected flights in three observational phases: Early winter: PGS5 – 21 December 2015, Mid–winter: PGS7 – 18 January 2016 and PGS12 – 31 January 2016, Late winter: PGS14 – 26 February 2016 and PGS19a – 13 March 2016.




Figure 7a summarizes all concurrent $NO_y$–$N_2O$ observations during POLSTRACC. The data are grouped in sub–datasets according to the three mission phases. The $NO_y$–$N_2O$ data field seems to be split into two major branches divided by the line representing $NO_y*$. During the early winter phase (points in black) the $NO_y$–$N_2O$ data pairs are essentially grouped around the $NO_y*$–curve. The mid–winter period (points in red) is mostly characterized by values lying above $NO_y*$, for $N_2O$

concentrations between about 250 and 300 ppb. The late winter (points in blue) shows values above $NO_y*$ as well as values below. Values below $NO_y*$ are associated with $N_2O$ values below about 250 ppb. As pointed out earlier, this cannot be attributed to the deviation of $NO_y$ from the linear correlation at low $N_2O$ concentrations which is only expected for $N_2O$ concentrations below about 100 ppb (Loewenstein et al., 1993). In Figure 7b, $dNO_y$ versus $N_2O$ is presented for the three mission phases. The values range between about +6 ppb and -6 ppb. In both figures, high values of $NO_y$ and $dNO_y$ for $N_2O$

values close to 320 ppb indicate tropospheric airmasses where the relation between $NO_y$ and $N_2O$ is not valid.

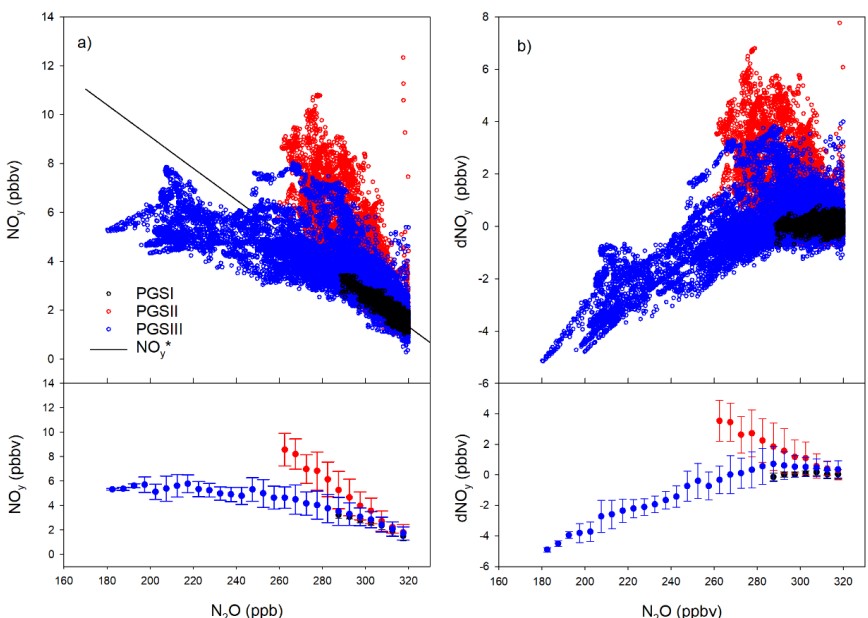

**Figure 7. (a) $NO_y$ and $NOy*$ versus $N_2O$ for all POLSTRACC flights. The three phases of the POLSTRACC mission (pre–winter, mid–winter, late–winter) are color–coded. Values are additionally given as means (averaged over 5–ppb $N_2O$ intervals) along with standard deviation. (b) Same as a) but $dNO_y$ versus $N_2O$.**


The influence of downward transport on nitrification or denitrification is also reflected in Figure 8. Here $dNO_y$ is shown versus potential temperature. Positive values of $dNO_y$ are predominantly found between 340 and 370 K during the flights in January. The largest denitrification was observed in air masses with potential temperatures between 390 and 410 K in late winter. The highest values of nitrification or denitrification, respectively were each found at the highest flight altitude of

approximately 14 km. During previous observations evidence for denitrification has mainly been found at higher altitudes in the stratosphere. E.g., in February 1995, measurements with the MIPAS–B balloon–instrument found a 50 % reduction in $NO_y$ at altitudes between 16 and 22 km (Waibel et al., 1999). Denitrification was also detected by satellite observations in winter 2009/2010 in the Arctic between about 475 and 525 K (Khosrawi et al., 2011). With the CLaMS model denitrification at about 500 K and nitrification at about 400 K was simulated for this winter (Grooß et al., 2014).





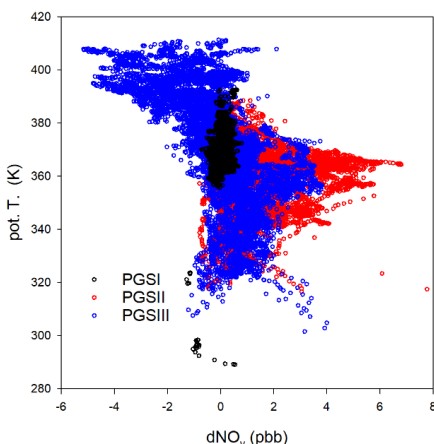


**Figure 8. dNO$_y$ color–coded for the three POLSTRACC–phases (early-, mid- and late winter) versus potential temperature.**

The POLSTRACC mission covered the whole winter season from December to mid of March providing the unique opportunity to probe the lowermost stratosphere under different conditions. The distribution of reactive nitrogen changed from undisturbed condition in early winter to a condition with elevated concentrations (nitrification) and finally to a

condition with lowered concentrations (denitrification) in late winter. This transformation of the reactive nitrogen distribution in the lower Arctic stratosphere is shown in Figure 6. In this figure the NO$_y$–N$_2$O relation for five selected flights is depicted. The sequence of these figures illustrates the temporal evolution of NO$_y$ at the bottom of the vortex. In early winter (6a) NO$_y$ and N$_2$O are well correlated reflecting that the distribution is controlled by the gas–phase production of NO$_y$ from N$_2$O. In mid–winter (6b and 6c) the observed NO$_y$ exceeds the calculated NO$_y$* by several ppb indicating the influence

of the evaporation of sedimenting particles containing nitric acid. In late winter the distribution of NO$_y$ is controlled by the downward transport of air masses that have undergone removal of nitric acid by heterogeneous processes (Figure 6e). Figure 6d shows a flight typical for the transition between the mid– and late winter distribution of NO$_y$. The observations at HALO flight altitude reflect the processes in the stratosphere above. Denitrification in the middle stratosphere by heterogeneous processes and removal by sedimenting particles happened at nearly the same time as nitrification by evaporating was

observed at the lowermost stratosphere. However, descending denitrified air masses at flight altitudes were first observed with a time lag of several weeks in late winter. In a certain way, the vertical distribution of NO$_y$ in the winter stratosphere was mapped to a temporal variation at HALO flight altitude.

As was pointed out earlier the NO$_y$/O$_3$ ratio is a further diagnostic tool for studying processes in the lower stratosphere. Evaporating particles lead to a nitrification of the lowermost stratosphere and this process finds its echo in the increase of the

NO$_y$/O$_3$ ratio as can be seen in Figure 2. The influence of the denitrification in the late winter phase does not form such an obvious signature in the NO$_y$/O$_3$ ratio. For example, Figure 5c shows the NO$_y$/O$_3$ ratio versus time for the flight on 13 March. Before about 37000 UTC s NO$_y$ and NO$_y$* agree reasonably well. There are no indications for denitrification during this part of the flight. In the further course of the flight the difference between NO$_y$ and NO$_y$* increased along with decreasing N$_2$O concentrations, a clear signature of denitrification. The NO$_y$/O$_3$ ratio, however, does not decrease during this

period as one might expect but it increased from about $3.9*10^{-3}$ to about $4.8*10^{-3}$. The obvious explanation for these observations is that not only NO$_y$ was removed from the air masses but also ozone. In sense of the NO$_y$/O$_3$ ratio the decrease in NO$_y$ by denitrification is counterbalanced by the decrease in ozone.

As a plausibility check, with a rough "back of the envelope" calculation the missing ozone can be estimated for this flight. Between about 37000 and 42200 UTC s, the denitrification amounted up to about 5 ppb, the NO$_y$/O$_3$ ratio increased to about





0.0048. Based on these observations and assuming an "undisturbed" $NO_y/O_3$ ratio of 0.004, on average in the order of about 1 ppm ozone were missing during this flight period. This is at least roughly in accordance with a much more detailed simulation using the CLaMS model. For this POLSTRACC flight a chemical ozone depletion of more than 1 ppm or about 50 % was estimated. (Oelhaf et al., 2019).

The temporal evolution of the subvortex region is also visible in an ozone–nitrous oxide coordinate system. Figure 9 shows

the correlation between these two trace gases for the same flights as in Figure 6. In addition, the regression line resulting from a linear least squares fit analysis of the December flight is plotted. Similar criteria apply to this tracer relationship as for the $NO_y$–$N_2O$ correlation (Hegglin et al., 2006; Hegglin and Shepherd, 2007; Bönisch, 2012). A slope of -18.5 was derived from the December 2015 flight. This compares quite well with the mid–latitude slope derived from the September data during the TACTS flights (see above) of -19.2. From late January through the end of the mission, the observed ozone

concentrations deviate more and more from the regression line to lower values. The difference increases with decreasing nitrous oxide concentrations, parallel to the evolution of the $NO_y$–$N_2O$ correlation in Figure 6. Thus, the denitrification observed in the descending air masses is also reflected in an ozone decrease.

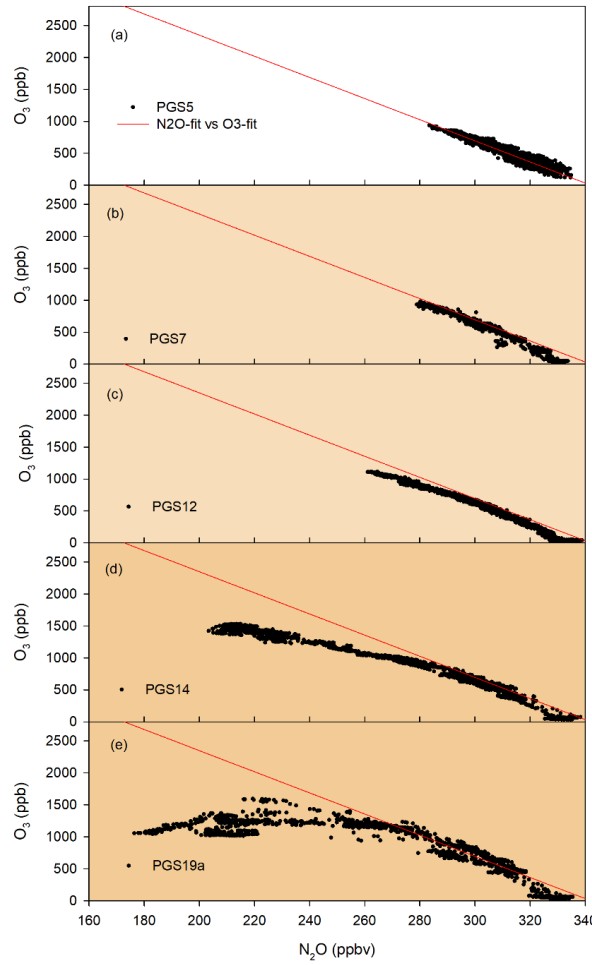

**Figure 9.** $O_3$ versus $N_2O$ for selected flights (same as in Figure 6) in three observational phases: Early winter: PGS5 – 21 December
**2015, mid–winter: PGS7 – 18 January 2016 and PGS12 – 31 January 2016, Late winter: PGS14 – 26 February 2016 and PGS19a – 13 March 2016. Additionally, the regression line resulting from the December flight is given.**



### 3.3 Observations of Particulate Nitrate

The POLSTRACC payload mainly comprised in situ gas–phase and remote sensing instruments. It did not include instruments for specific measurements of particle parameters. However, the $NO_y$ detector offers an indirect method to

observe particles containing reactive nitrogen compounds. For these measurements the oversampling characteristic of the forward–facing inlet was used. With the backward facing inlet, only gas–phase total reactive nitrogen is measured (see Section 2.1.1). Particles evaporating within the inlet–system release nitrate molecules that are detected by the $NO_y$ instrument. During one episode during the flight on 18 January, the resulting chemiluminescence signal was even so high that it could not be processed by the detection electronics.

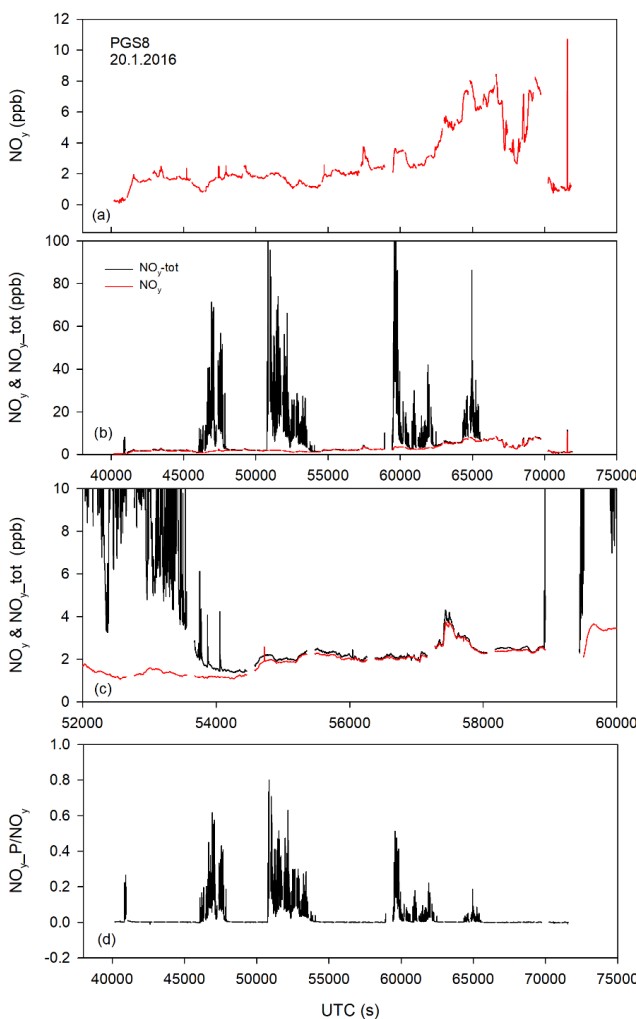

**Figure 10. POLSTRACC–flight on 20 January 2016. During this flight particulate nitrate was observed. (a) Gas–phase $NO_y$, (b) Total nitrate (including particulate nitrate) and gas–phase nitrate, (c) same as (b) but with a smaller time interval, (d) Ratio of particulate nitrate (corrected for enhancement) to gas–phase nitrate.**

During the midwinter phase of POLSTRACC particulate nitrate has been observed during four flights using this

measurement approach. The flight on 20 January went from Kiruna north towards Spitzbergen, then westwards towards Iceland and back to Kiruna. One mission objective was to probe forecasted nitrification of the lowermost stratosphere. Gas–phase $NO_y$ ranged between about 1 and 8 ppb at altitudes between 12.5 and 14.3 km (see Figure 10). During four periods the





signal obtained with the forward–facing inlet was up to a factor of 50 higher than the gas–phase signal obtained with back–ward facing inlet. In Figure 10b total $NO_y$ measured with the forward–facing inlet is shown along with gas–phase $NO_y$.

$NO_y$–tot is not corrected for enhancement and shows values up to 100 ppb. In total, the areas in which particle $NO_y$ was observed during this flight extended over about 2000 km. Outside these episodes $NO_y$ obtained with the forward and aft facing inlet, respectively, agreed within about 7 % percent.

Particulate nitrate was observed in situ on HALO only during the mid–winter phase. In addition to the 20 January flight, particulate nitrate was also observed during the flights on 18, 25 and 31 January. The large–scale particulate events were

observed over distances of approximately 800 to 1400 km. Widespread polar stratospheric clouds of different compositions have also been observed with the WALES lidar instrument (Water Vapour Lidar Experiment in Space – airborne demonstrator) on board the HALO aircraft above flight altitude (Voigt et al., 2018). Between December and end of January PSC particles have been detected with space–borne LIDAR between 15 and 26 km (Pitts et al., 2018; Voigt et al., 2018).

The measurement approach, using the oversampling characteristic of inlets has also been used during earlier aircraft

missions in the Arctic. High $NO_y$–net values indicating particulate nitrate have been observed during the SOLVE mission in winter 1999/2000 from the ER–2. Large fields of PSC particles were found between about 15 and 21 km (Northway et al., 2002). Also balloon–borne measurements confirmed the presence of NAT (Voigt et al., 2000) in that winter while liquid ternary solution particles (Schreiner et al., 1999) were observed in the THESEO winter before. During the Vintersol/Euplex mission in February 2003 single NAT particles have been observed at altitudes between about 18 and 20 km onboard the

Geophysica (Voigt et al., 2005). Compared to previous observations, during POLSTRACC particulate nitrate was found at lower altitudes between about 10 and 14.5 km. This corresponds roughly to potential temperatures between about 310 and 370 K (Figure 11).

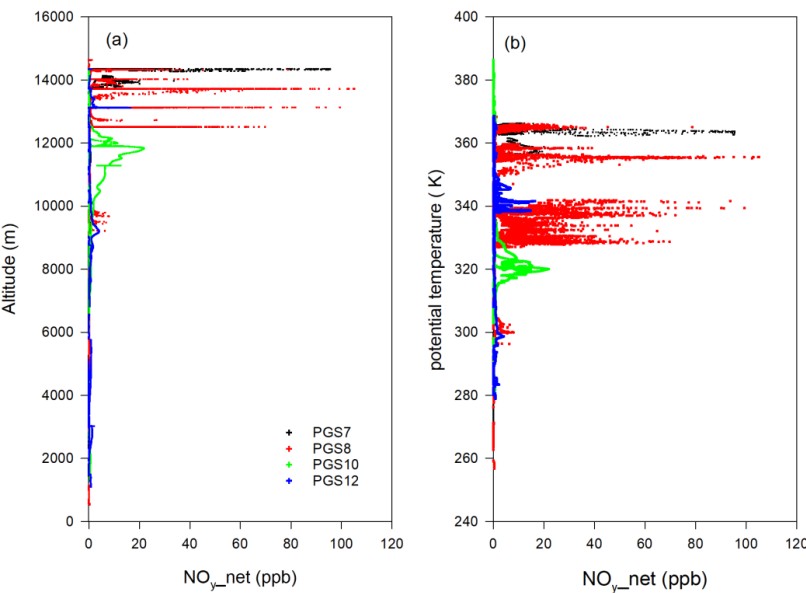

**Figure 11. $NO_y$–net versus altitude (a) and $NO_y$–net versus potential temperature (b) for all flights with the occurrence of**
**particulate nitrate (18, 20, 25, 31 January 2016).**

The vapour pressure of nitric acid over NAT can be derived from measured $NO_y$, water vapour, ambient pressure and temperature using the expression given by Hanson and Mauersberger (1988). For estimating the saturation ratio one can assume that observed $NO_y$ mainly consists of nitric acid in the stratosphere as confirmed by simultaneous measurements of





nitric acid (Figure 1). In Figure 12 the saturation ratio of nitric acid is plotted along with the gas–phase equivalent of particulate nitrate. In general, a good agreement between the occurrence of particulate nitrate and high NAT saturation ratios was found. During the flight on 18 January, the saturation ratio was up to about 0.25 indicating ongoing evaporation of the particles. During the particle observations of the other flights the air masses have been supersaturated with respect to NAT.

To convert total $NO_y$ to an equivalent gas–phase $NO_y$ concentration, the particle size dependent enhancement factor has to be

known. As mentioned earlier, the HALO payload did not include instruments for the independent measurement of the particle size. The enhancement factor therefore can only be estimated with some uncertainty from the signal obtained with the forward–facing inlet. As pointed out earlier, nitrate containing particles evaporate completely in the inlet and converter and the released nitrate molecules are detected. From this signal the diameter of the particles can be derived. This method was described in detail by Northway et al. (2002). Due to wall effects particles do not evaporate within the sampling

frequency of the instrument (1 second). The complete evaporation takes up to 20 s. Therefore, for estimating the diameter of the particle, the signal has to be integrated over the complete time of evaporation. Assuming NAT density and for the specific parameters of the AENEAS $NO_y$–detector the particle diameter can be estimated. The pre–factor of the formula combines density and specific instrument parameters. The measured concentration enters the formula with the units mol/mol:

$D[m]=4.28*10^{-3}(NO_y)^{1/3}$  (3)

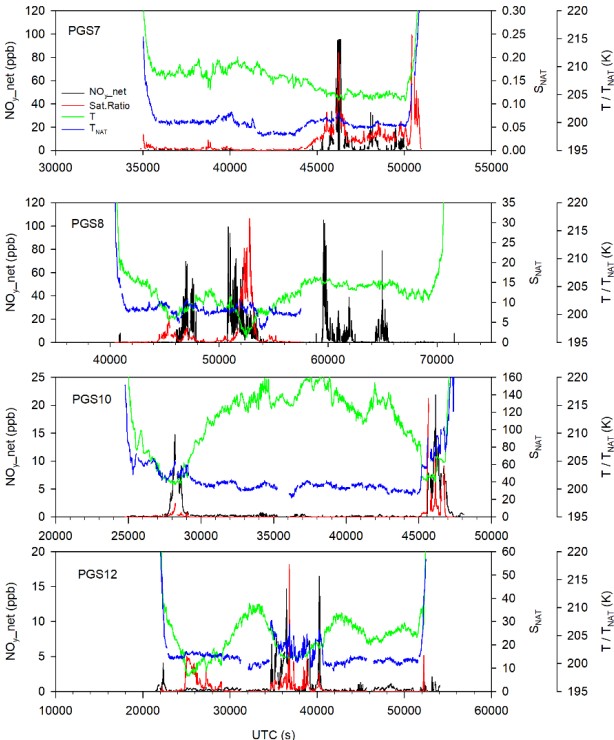

**Figure 12. $NO_y$–net, ambient temperature, saturation ratio and $T_{NAT}$ for all flights where particulate nitrate was observed. During the second half of the flight PGS8 the NAT saturation temperature $T_{NAT}$ could not be calculated because of missing data.**

A similar approach was also chosen for analysing particles from observations from the Geophysica (Voigt et al., 2005). As

was shown by Northway et al. (2002) this method can only be applied as long as the evaporation periods of the individual particles do not overlap. During POLSTRACC the frequency of evaporating particles was that high that for most time this approach could not be used to estimate the particle diameter. At the edge of some particle episodes, however, individual evaporation events could be identified. Using the above given formula, particle diameters between about 9 and 18 µm have

been derived. The particle diameters derived from the POLSTRACC observations are roughly in the same order of magnitude as those measured during previous aircraft missions to the Arctic. Particle diameters, between 5 and 20 µm, have been derived from measurements with the ER–2 during the SOLVE mission at altitudes between 15 and 21 km (Northway et al., 2002; Fahey et al., 2001). During RECONCILE in January 2010, particle diameters between 10 and 24 µm have been concluded from observations on board of the research aircraft Geophysica (Molleker et al., 2014). Particles with diameter smaller than 6 µm have been observed from the Geophysica in February 2003 (Voigt et al., 2005).

As discussed in Section 2.1.1, the enhancement factor reaches a maximum for particles larger than about 10 µm and is then independent of diameter (Belyaev and Levin, 1974). It also increases strongly with increasing ambient pressure. The maximum enhancement factor ranged between about 50 and 85 for the instrumental setting during this mission at the altitudes where particles were detected. For comparison, an enhancement factor between about 13 and 22 was applied for measurements on board of the ER–2 at lower ambient pressures during AAOE in the Antarctic (Fahey et al., 1989). For observations during POLSTAR at altitudes up to 13 km a maximum enhancement factor of 140 was derived for particles larger than 10–20 µm (Feigl et al., 1999).

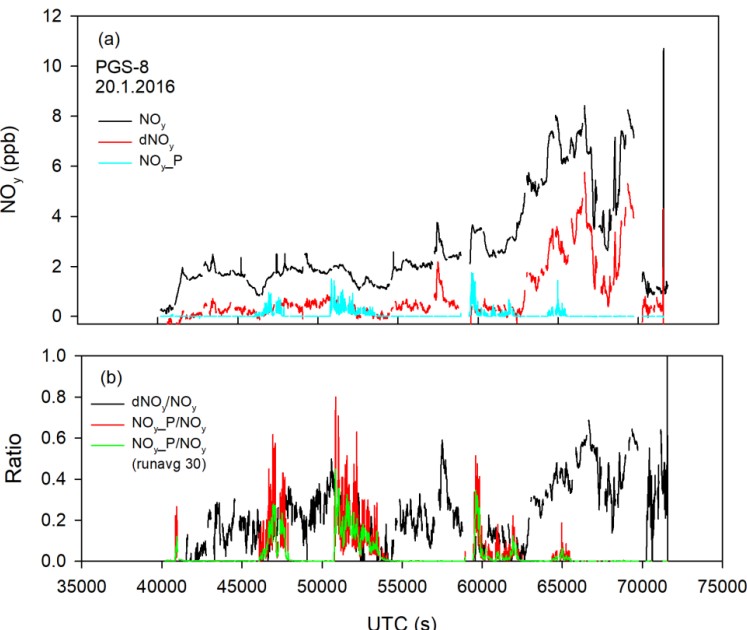

**Figure 13.** $NO_y$–partitioning for PGS–flight 8. Gas–phase $NO_y$, $dNO_y = NO_y - NO_y^*$ (calculated) and enhancement corrected particulate nitrate are shown. (a) absolute values, (b) $dNO_y/NO_y$ ratio, $NO_y\_P/NO_y$ ratio – 1 s values and as running average (30s).

Following the above sketched assumptions, the gas–phase equivalent of the observed particulate nitrate can be estimated as is presented for the observations during the flight on 20 January (Figure 13). This allows to establish a kind of $NO_y$–partitioning. The observed total reactive nitrogen in the lower stratosphere comprises three contributions: a) gas–phase $NO_y$ arising from the photo–oxidation of $N_2O$ – "undisturbed $NO_y^*$", b) $NO_y$ from already evaporated nitrate particles and c) particulate nitrate. For the flight on 20 January observed $NO_y$ exceeds calculated $NO_y^*$ by up to 6 ppb. During the last part of

this flight up to about 60 % of observed $NO_y$ can be attributed to evaporated particles. Averaging over the whole flight, the median ratio between excess $dNO_y$ and $NO_y$ was about 20 %. The highest values for particulate nitrate amounted about 1.5 ppb. The highest peak ratio between particulate nitrate and gas–phase nitrate was up to more than 0.7. Averaged over a longer period (50800 to 53500 s UTC) particulate nitrate amounted around 0.2 ppb. This corresponds to a ratio between particulate and gas–phase nitrate of about 0.14. High $dNO_y$ values were not necessarily observed at the same time as high





particulate nitrate, indicating that the particles have been already completely evaporated. With the exception of the peak

values, more reactive nitrogen was found in the gas–phase than in the particulate phase.

Besides the above sketched dependences, the estimation of the enhancement factor might be hampered by the fact that the aircraft itself has an influence on the sampling characteristic of the inlets on the top of the fuselage. This was shown in a publication studying the influence of the inlet position on the quantitative determination of the ice water content. (Afchine et

al., 2018). Depending on size of the ice particles, overestimation as well as underestimation of the ice water content was found for the specific inlet position of the FISH instrument. Due to the different positions of the water and reactive nitrogen inlet and the different particle size distributions of the NAT PSC and cirrus ice crystals these finding cannot easily be transferred to the present measurements. However, an influence of the aircraft on the particle sampling cannot be excluded although this effect might be smaller for the smaller NAT particles compared to ice. This might affect the enhancement

factor used for the estimation of the gas–phase equivalent of particulate nitrate and therefore might affect the quantitative determination of the ratio between particulate and gas–phase nitrate. However, this uncertainty does neither affect the fact that nitrate particles have been observed nor does it affect the estimate of the particle diameter. It also has no influence on the gas–phase measurements of nitrified and denitrified air masses.

**4. Model simulations**

The distribution of reactive nitrogen in the lowermost winter Arctic stratosphere was also simulated with the Chemical Lagrangian Model of the Stratosphere (CLaMS). CLaMS was developed at the Forschungszentrum Jülich as modular chemistry transport model. With CLaMS, denitrification in the Arctic stratosphere in winter 2003/2003 and 2009/2010 was simulated and compared to in situ measurements from the Geophysica (Grooß et al., 2005; Grooß et al., 2014).

The simulation of the impact of vertical settling of NAT particles is challenging as it depends critically on temperature and

also the heterogeneous nucleation (Tritscher et al., 2021). For example, nitrification occurs when the sedimentation of NAT particles fall into altitudes with temperatures above $T_{NAT}$. This results in a horizontally and vertically filamentary small–scale structure in the $NO_y$ distribution, seen both in the observations during the campaign but also in the simulation. As time proceeds, these $NO_y$–rich structures are diluted and mixed with ambient air. Due to the limited knowledge of NAT formation nuclei, small–scale temperature distributions and also due to the resolution of the model, an exact and detailed simulation of

the nitrification filaments is not possible. In the simulation, all of the $HNO_3$ released from NAT particle evaporation is collected in the air parcels that have about 100 km distance. In some cases, the interpolation of the resulting simulated $NO_y$ to a specific location as the flight path may result in unrealistically high values.

First, we compare in situ $NO_y$ measurements with model values simulated along the flight paths. In Figure 14 measured and simulated values are presented for selected flights for the three winter phases. During the early winter phase, the lower

stratosphere does not show any signature for nitrification or denitrification as was shown in section 3.2.1. As an example, the flight from Oberpfaffenhofen to the Arctic on 21 December was chosen (Figure 14a). Up to that time, no vertical $NO_y$ redistribution was simulated by CLaMS and therefore $NO_y$ and $NO_y^*$ are identical. The larger structures of the observed in situ $NO_y$ variations along the flight path have been largely reproduced by the simulations. There are a few smaller–scale variations that are not met by the model, e.g. a larger deviation of about 1 ppb $NO_y$ in absolute numbers is found between

about 50000 and 55000 UTCs.





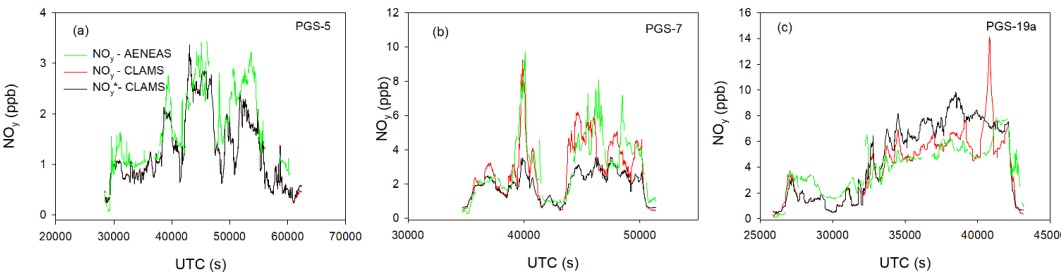

**Figure 14. Comparison between observed NO$_y$ and simulated NO$_y$ values with the CLaMS–model. Additionally, NO$_y$\* calculated with CLaMS is given. Shown are examples for three POLSTRACC flights from the three observation phases. Early winter: PGS–5 – 21 December 2015, Mid–winter: PGS7 – 18 January 2016, Late winter: PGS19a – 13 March 2016.**


For the midwinter phase, the flight on 18 January has been chosen for comparison (Figure 14b). During this flight, indications for particulate nitrate have been found by the AENEAS observations. Based on tracer–tracer correlation the air masses encountered have been affected by nitrification (Section 3.2.2). Besides simulated NO$_y$ also NO$_y$\*–CLaMS is shown. It corresponds to simulated NO$_y$ without considering heterogeneous reactions and subsequent redistribution.

In general, again model and measurement capture the same large–scale features. Both observations and simulations indicate an enhancement of NO$_y$ during the same times or locations of the flight path. Typical NO$_y$ enhancements are of the order of 2–3 ppb, but also the nitrification peak of 10 ppb NO$_y$ was reproduced by the model. Deviations can be found at a smaller scale where the nitirification patterns are not congruent at all times. During episodes that are not affected by nitrification deviations between simulation and observation are below 1 ppb.

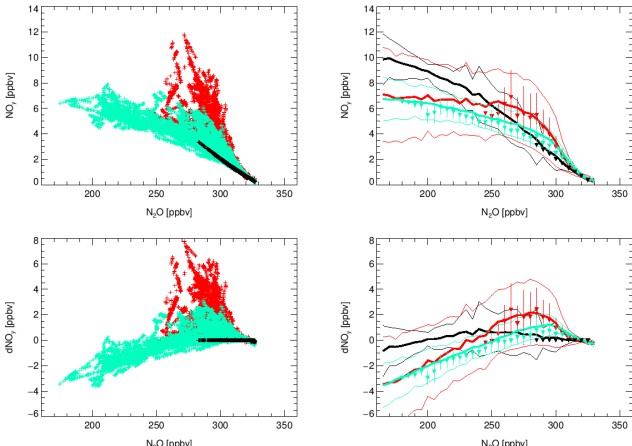


**Figure 15. Correlations of NO$_y$ (top panels) and dNO$_y$ (bottom panels) with N$_2$O from CLaMS simulations. The left panels show all date in different colours for the three campaign phases. Data within 5 minutes of the times 31.1.16/14:17 and 13.3.2016/11:20 UTC are omitted (see text). The right panels show the averages for N$_2$O bins +–1sigma standard deviations of these values as filled circles with error bar signs. Also shown are the vortex average correlation for central dates of the campaign phases as thick**
**coloured lines, +–1sigma standard deviation is indicated as thin coloured lines.**

As example for the late winter period the 13 March flight was chosen. The air masses probed in the March flight show indications of denitrification over longer periods. Both, observations and model simulations reveal the patchy filamented structure of the lower stratosphere. The simulation indicates that denitrification reveals a NO$_y$ reduction by 2–4 ppb
throughout the second half of the flight. Also, in this case, the NO$_y$ deviations between the simulation and observations are generally around 1 ppb with a few exceptions. Around 11:20 UTC (40800 s), the simulation indicates a strong nitrification



peak of about 14 ppb $NO_y$ that is not present in the observations. A detailed investigation revealed that this nitrification filament in the simulation belongs to one short episode on this flight west of the coast from Greenland (lon 303.4, lat 71.7, theta 404.1K) where the flight path is very close to one specific air model air parcel (distance 7 km). In this air parcel all

evaporating $HNO_3$ is collected over the range given by the horizontal model resolution of about 100 km. Small uncertainties in the temperature of the reanalyses could cause a small displacement of such a filament. It may be that there is a similar nitrification filament close to the flight path but this cannot be clarified here.

Besides these comparisons along individual flight paths, we used the CLaMS simulation to investigate if the vertical $NO_y$ redistribution on a vortex–wide scale can be understood. This is done by examining the correlation on $NO_y$ and $dNO_y$ with

the inert tracer $N_2O$. The simulations were used to put the observations in a broader context. Especially they can contribute, to what extent the observations represent the global development within the polar vortex. To do that, the simulations were evaluated as vortex average and also interpolated to the HALO flight paths. For comparing the time development of denitrification and nitrification in the three campaign phases, we evaluated the model averages at central dates for each of the campaign phases (17 December, 22 January, and 8 March). Similar to the observations combined in Figure 7, Figure 15

shows the correlation of $NO_y$ and $dNO_y$ with $N_2O$ as interpolated to the flight path and time obtained with CLaMS. For the comparison, two short 10–minute periods out of the over 250 flight hours were excluded (within 5 minutes of the times 31.1.2016/14:17 and 13.3.2016/11:20 UTC) that they were close to a single model air parcel with likely an over–estimation of $NO_y$ as described above. The right panels of Figure 15 show two kinds of averages for $N_2O$ bins: triangles show the average $NO_y$ as interpolated to the flight path position and thick coloured lines show the average vortex $NO_y$ mixing ratios.

The error bars and the thin coloured lines indicate the +/- 1σ range. As these two evaluations, vortex average and average at the flight path, overlap well, we conclude that the ensemble of all observations are representative for the vortex–wide vertical $NO_y$ redistribution.

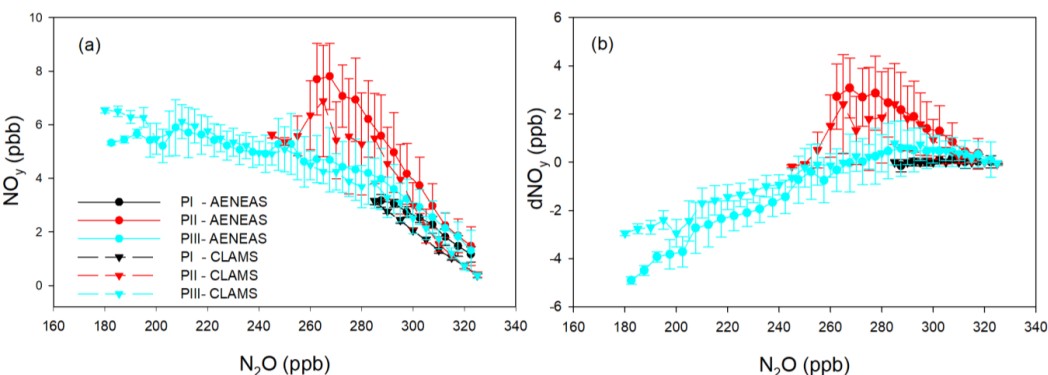

**Figure 16. (a) $NO_y$ and (b) $dNO_y$ averaged over 5–ppb $N_2O$ intervals along with standard deviations for the three mission phases.**
**Observed values (circles) are shown along with values from the CLaMS–model simulation (triangles).**

Finally, Figure 16 combines observed and simulated (along the flight paths) $NO_y$ and $dNO_y$ correlation with $N_2O$. In general, observations and simulations agree well with deviations below about 1 ppb. Only few measurement points were obtained below $N_2O$ values of 200 ppb, which could explain the larger discrepancy between simulated and observed values.

The comparison between observations and model simulations is a multi–step process. First, we were able to show, through the comparisons along the flight paths, that the processes underlying the model simulations are so well understood that the observations can be reproduced. In a second step, we showed that these simulations along the flight path are representative for the vortex–wide $NO_y$ distribution. This, in turn, suggests that the observations, although limited in time and space to individual flights, provide a good description of the distribution of nitrogen oxides in the subpolar region during this winter.



## 5 Summary


During the course of the extremely cold winter 2015/2016 aircraft–based measurements with the German research aircraft HALO have been performed in the lowermost stratosphere of the Arctic region within the POLSTRACC mission. The observation period covered the whole winter–spring season from December to March. This extended observational period offered the unique opportunity to study the changing distribution of total reactive nitrogen in the lowermost stratosphere with 685 time.

Tracer–tracer correlations, the relation between $NO_y$ and $N_2O$ and $O_3$, respectively have been used as tool to study and interpret the observed temporal evolution of the UTLS composition. In December, the distribution of reactive nitrogen did not show any indications for deviations from undisturbed conditions controlled by gas–phase chemistry and transport. This changed during the second mission phase in January and beginning of February. During several flights enhanced $NO_y$ values 690 have been observed. Using $NO_y$–$N_2O$ tracer correlations, nitrification could be clearly identified. Observed $NO_y$ values have been up to 6 ppb higher than expected without redistribution of nitrogen species. This is also reflected in the $NO_y/O_3$ ratio that was up to more than a factor of two higher in January than in December. This could be interpreted in terms that the sub–vortex region was affected by heterogeneous processes taking place in the stratosphere above. Particles falling down from the PSC regions evaporate and released gas–phase $NO_y$ leading to a nitrification of the lowermost stratosphere. This means, 695 that during some periods more than 60 % of the observed $NO_y$ was caused by evaporating particles.

Along with enhanced gas–phase values particulate nitrate was observed during mid–winter at flight altitude between about 10 and 14 km. The occurrence of PSC particles at such altitudes is rare. Particulate nitrate was observed over wide regions during four flights out of Kiruna. The diameter of these PSC particles ranged between about 9 and 18 µm. The occurrence of particulate nitrate at flight altitude and the nitrification of the lowermost stratosphere fit into the picture of other observations 700 made in this winter. Extended PSC coverage has been already observed during December by CALIOP (Pitts et al., 2018). Particulate nitrate formed in the middle stratosphere was also observed by LIDAR on board of HALO and simulated by models (Voigt et al., 2018; Khosrawi et al., 2017). However, the uncertainty of the sampling characteristic of the inlet makes a quantitative determination of the ratio between particulate and gas–phase nitrate difficult.

Nitrified regions at the lowermost stratosphere have also been found in the late winter phase. Along with nitrified regions 705 subsidence of air masses from the polar vortex controlled more and more the distribution of reactive nitrogen at flight altitudes. Using tracer–tracer correlations substantial denitrification could be derived in subsiding air masses with minimum values of down to about -5 ppb. This means that up to about 50 % of the undisturbed $NO_y$ was missing.

Nitrification of the lowermost stratosphere in mid–winter and denitrification in late winter are linked together by heterogeneous processes in the above lying stratosphere. While nitrification caused by sedimenting particles was already 710 observed in mid–winter at flight altitude, the result of the denitrification at higher altitudes was not observed at the bottom of the vortex before end of February. Concurrently with dentrification, lower ozone concentrations were observed in the sinking air masses indicating ozone destruction at higher altitudes.

In situ observed total reactive nitrogen has been compared with the results of simulations with the Chemical Lagrangian Model of the Stratosphere (CLaMS). In general, CLaMS simulations reproduced the observed overall $NO_y$ structures and 715 concentrations. This is true for undisturbed conditions in December as well as for nitrified conditions in January and denitrified conditions in February and March. The comparison with the model simulations suggests that the observations during POLSTRACC have been representative for the vortex–wide vertical $NO_y$ redistribution.

Thus, the present measurements provide a comprehensive picture of the temporal evolution of the reactive nitrogen distribution during a whole winter period. They allowed to observe the transition of the lowermost sub–vortex region from 720 the undisturbed to the nitrified and finally denitrified state.



**Data availability:** The observational data obtained during the HALO flights are available at the HALO database: https://halo-db.pa.op.dlr.de.

**Author contributions:** HZ, GS, PS and ML were responsible for the $NO_y$ measurements. HZ for the further data analysis and the writing of the manuscript. PH and JK performed the $N_2O$ measurements. JUG performed the CLaMS model simulations. AZ was responsible for the Ozone measurements. AA and CR carried out the water observations. Airmass age was provided by AE, $HNO_3$–measurements with the AIMS mass–spectrometer were performed by AM and CV. $HNO_3$ observations by GLORIA were provided by WW, MB, JU, and the GLORIA Team. HO and BMS were responsible for the
scientific flight planning and GLORIA observations. All co–authors were involved in review and editing of the paper.

**Competing interests**: The authors declare that they have no conflict of interest.

**Acknowledgements.** We would like to thank all colleagues of the DLR Flight Department for their excellent support, which
made this successful flight campaign possible. We thank the entire POLSTRACC-GW-LCYCLE-SALSA team for the very productive and pleasant collaboration. We thank all colleagues who participated in the meteorological flight planning. The authors would like to thank the Earth System Modeling (ESM) project for funding the CLaMS simulations by providing computation time on the ESM partition of the JUWELS supercomputer. We thank Martin Dameris (internal review) for many helpful comments.


**Financial support.** We gratefully acknowledge funding from the German Research Foundation (DFG) under the Priority Program "Atmospheric and Earth System Research with the High Altitude and Long Range Research Aircraft (HALO)" SPP 1294 through the following grants: Andreas Engel and Peter Hoor (EN 367/13-1, EN 367/14-1, HO 4225/7-1). Jens Krause was partially funded by grant HO4225-6/1. Andreas Marsing and Christiane Voigt were funded by the DFG SPP HALO
1294 under contract numbers VO1504/6-1, VO1504/7-1 and by the Helmholtz Association in the HGF-W2/W3 excellence program.




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
