# Peer review of "Redistribution of total reactive nitrogen in the lowermost Arctic stratosphere during the cold winter 2015/2016"

_Atmospheric Chemistry and Physics, 2021_

## Author Response (AR1)

Acp-2021-707

Helmut Ziereis

Redistribution of total reactive nitrogen in the lowermost Arctic stratosphere during the cold winter 2015/2016.

**This document contains a list of all changes made to the manuscript. It also contains the responses to the reviewers' comments.**

The line numbers of the listed changes to the original manuscript (9/22/2021) refer to the track version of the manuscript.

Most changes were initiated by reviewer comments and are also listed in the responses to reviewers.

| Line / Figure | Comment | Changed text |
|---|---|---|
| | | |
| 7 | New author included in the list | Vera Bense |
| 21-41 | Following the suggestions of the reviewer the abstract was rebuild | |
| 22 | | … with a very strong polar vortex and … |
| 29-33 | | … correlations ($NO_y$-$N_2O$ and $NO_y$-$O_3$). The trace gases are well correlated as long as the $NO_y$ distribution is controlled by its gas-phase production from $N_2O$. Deviations of the observed $NO_y$ from this correlation indicate the influence of heterogeneous processes. In early winter no such deviations have been observed. In January, however, air masses with extensive nitrification were encountered at altitudes between 12 and 15 km … |
| 37 | | … at lower altitudes … |
| 40 | This line has been moved | Using tracer–tracer correlations, missing total reactive nitrogen was estimated to amount up to 6 ppb. |
| 41-43 | Following the comments of the reviewer a new analysis and figure 9 was included in the manuscript. The point addressed in the new analysis is reflected also in the abstract. | Further, indications of transport and mixing of these processed air masses outside the vortex have been found, contributing to the chemical budget of the winter lower most stratosphere. |
| 60 | The line was changed | … supply … |
| 61 | The line was changed | … by … |
| 74 | UTLS: the abbreviation was explained | Upper Troposphere – Lower Stratosphere |
| 108 | The line was changed | … following … |
| 123 | Table 1 was added | |
| 129 | | … and the whole mission… |
| 188 | This sentence was added following the suggestion of a co-author. | The FISH instrument was connected to a forward-facing inlet and thus can detect ice |

| | | particles (evaporated at the inlet walls) in addition to the gas-phase water vapor (Afchine et al., 2018) |
|---|---|---|
| 226 | These sentences have been added to meet the comments of the reviewer. | The lifetime of nitrous oxide is more than 100 years (Prather et al., 2015). The mean age of an air parcel can be understood as average time since the last contact with the troposphere (Ploeger et al, 2015). During POLSTRACC the mean age of the probed air masses ranged between about 1 and 5 years (Krause et al., 2018). |
| 236 | | … (Murphy et al., 1993). |
| 253 | The sentence has been changed to reflect the comments of the reviewer and to clarify the point addressed. . | As long as there are no additional processes, sources or sinks, in the lower stratosphere affecting the $NO_y$ concentration, observed $NO_y$ should be very close to $NO_y$* (within the uncertainty range of observations). |
| 268 | The sentence has been removed. The related statement is addressed elsewhere in the manuscript (line 282 ff). | |
| Figure 1 | Following the suggestion of the reviewer a uncertainty range was included in Figure (b) | The uncertainty range arising from the calculation of $NO_y$* is shaded in grey. |
| 283-290 | Following the comments of the reviewer, this paragraph was rewritten. | … Also included in this figure is the regression line resulting from a linear least squares fit ($R^2$=0.87). The range of its uncertainty is indicated by dashed lines. As expected for undisturbed conditions, $NO_y$ and $N_2O$ are anticorrelated. To exclude tropospheric values that would affect the correlation, only values obtained in the stratosphere have been used for this analysis. In 2016 the tropospheric $N_2O$ concentration amounted about 329 ppb (Combined Nitrous Oxide data from the NOAA Global Monitoring Laboratory). Therefore, the analysis was performed only for $N_2O$ values smaller than 320 ppb. The slope of the regression line, corresponding to the factor f given in Eq. (3), is about 0.064. This value … |
| 294 | | … obtained during the TACTS mission … |
| 298 | | The equation describing the regression can be rewritten to take the form of Eq. (3). In this formulation, the following calculations of NOy* were performed. |
| 300 | | The slope obtained during the midlatitude mission TACTS was chosen as conversion efficiency f. |

| | | |
|---|---|---|
| 312-314 | Following the comments of the reviewer, these sentences were rewritten. | … tropopause where the relative contribution of tropospheric $NO_y$ to $NO_y*$ is largest. With decreasing $N_2O$ concentration and increasing stratospheric character of the air mass $NO_y$ arising from the photooxidation of $N_2O$ increases… |
| 322 | | In Figure 1b measured $NO_y$ values are shown along with calculated $NO_y*$ values. Also shown is the uncertainty range of NOy*. |
| 344-345 | Explanation of PV | potential vorticity values of more than 2 PVU. The height of the dynamical tropopause is commonly attributed to the level where the potential vorticity equals this value. |
| 347 | | … in 2012. |
| 350-351 | | At altitudes above 12 km, significantly higher $NO_y$ concentrations were measured, with values up to about 10 ppb, than during the December flight, with maximum values up to 3.4 ppb (Fig. 2). |
| 360 | | … of the $NO_y/O_3$ ratio. |
| 407 | | The flight on 26 February (Figure 4 and 6d) may serve as an example |
| 427 | | On average, the difference is about 0.08 ppb with a standard deviation of about 0.48 ppb. |
| Figure 6 | 6 (a) now includes the uncertainty range for NOy* as suggested by the reviewer. | … . The uncertainty range arising from the linear least squares fit for the PGS-flight 5 is indicated by dashed lines. … |
| Figure 6 | The caption was changed to standardize the nomenclature for the flights. | PGS-**flight** 5 – 21 December 2015, Mid–winter: PGS-**flight** 7 – 18 January 2016 and PGS-**flight** 12 – 31 January 2016, Late winter: PGS-**flight** 14 – 26 February 2016 and PGS-**flight** 19a – 13 March 201 |
| Figure 7 | Figure 7 now includes the regression line and its uncertainty range as suggested by the reviewer | Calculated $NO_y*$ is shown as a solid line, the uncertainty range is indicated by dashed lines. |
| Figure 9 | As suggested by the reviewer a new analysis was performed and its results now included as Figure 9 | **Figure 9.** Distribution of dNO_y in coordinates of equivalent latitude and potential temperature (theta) during phase I (13 December - 21 December), phase II (12 January - 2 February) and phase III (26 February - 18 March). The black contours show potential vorticity in PVU. |
| 499-511 | To explain Figure 9 the text was added. | Air masses processed in the polar vortex can also be transported to mid-latitudes. Figure 9 shows dNO_y in coordinates of equivalent latitude and theta (potential temperature). Equivalent latitude takes advantage of the adiabatically quasi-conserved nature of potential vorticity. It therefore removes the variability in trace gas distributions that originates from reversible deviations from the climatological mean due to Rossby and |

| | | |
|---|---|---|
| | | smaller scale waves (see e.g. Hegglin et al., 2006). The early-winter period shows a relatively undisturbed distribution of reactive nitrogen, the $dNO_y$ values are close to zero. The mid-winter period is mostly characterized by positive $dNO_y$ values, particularly above 340 K and polewards of 50$^\circ$ N equivalent latitude. The late winter period shows a nitrified region at the same location, but with weaker nitrification than in phase II. A denitrified region is located above, predominantly at potential temperature over 380 K and equivalent latitudes over 50$^\circ$ N. However, weak denitrification with losses up to 1 ppb is also observed throughout the whole latitude range above 360 K, even outside the vortex. Similarly, at lower isentropes slightly positive values of $dNO_y$ at lower equivalent latitudes are consistent with export of former vortex air to lower latitudes (Hoor et al., 2004, Krause et al., 2018). These findings indicate transport and mixing of vortex processed air masses to the mid-latitude lowermost stratosphere in late winter and early spring. |
| Figure 10 ff | Due to the inclusion of a new figure the number of all following numbers increased accordingly. | |
| Figure 10 | To be consistent with Figure 7: the uncertainty of the regression was added. | The uncertainty range of the regression is indicated by dashed lines. |
| Figure 12 | The green color of a line was replaced with cyan. | |
| Figure 13 | The green color of a line was replaced with cyan. | |
| Figure 14 | The green color of a line was replaced with cyan. | |
| 786 | New author contribution | VB performed part of the $N_2O$-analysis. |
| 807-810 | | DFG SFB/TR 301 TP-Change and by the Helmholtz Association in the HGF-W2/W3 excellence program. We gratefully acknowledge support by the SFB/TR 301 (TPChange: The Tropopause Region in a Changing Atmosphere, project no. 428312742) funded by the Deutsche Forschungsgemeinschaft (DFG, German Research Foundation). |
| 818ff | All references have been revised. This applies in particular to the doi specifications. Three new references have been included: Hoor et al., | |

| | | |
|---|---|---|
| | 2004; Ploeger et al., 2015; Prather et al., 2015. | |
| | | |
| | | |
| | | |
| | | |
| | | |

Answers to Comment on acp-2021-707
Anonymous Referee #1
16.12.2021

The referee's original comments are in *italics*. Our responses are written in plain black font. Changes to the manuscript text are shown in red.

We thank the referee for comments and suggestions that help to improve our manuscript.

**General comments**

*1) The discussion of tracer-tracer correlations (N2O – NOy) and in particular the comparison between NOy and NOy\* during the early phase of the campaign - before renitrification occurred - could be more quantitative. The results of a York-Fit (R2; slope (+- STD)) for the data in Figure 6a and Fig 7a could give a better understanding how accurate the relation between NOy and N2O is. In a similar way, a quantitative study on the deviations between NOy and NOy\* in Figure 1b would give an indication on the smallest amount of NOy change that can be derived from the data.*

**Answer:** To quantify the linear least squares fit between $NO_y$ and $N_2O$ we added the uncertainty range of the slope arising from the regression as dotted lines to Figure 6a and 7a  Also, I added the uncertainty range in Figure 1b as shaded area.

In parenthesis I added the value for $R^2$.

I added the following sentences to the text:

"… . Also included in this figure is the regression line resulting from a linear least squares fit ($R^2$=0.87). The

range of its uncertainty is indicated by dashed lines.  …"

…

"In Figure 1b measured $NO_y$ values are shown along with calculated $NO_y$\* values. Also shown is the uncertainty range of $NO_y$\*. During most of the time both curves agree well within the uncertainty range…"

*2) As mentioned in the manuscript, the individual flights covered a large area from the mid-latitudes to the northern sub-vortex region, with the majority of the observation made at high latitudes. It would interesting to see, whether signatures of re- and denitrification occur exclusively below the polar vortex, or whether vortex processed air-masses are transported to the mid-latitudes. This could be done e.g. by classifying air masses with deviations in NOy relative to the vortex edge (e.g. using equivalent latitude).*

**Answer:** We added a new figure, Figure 9, to the manuscript where $dNO_y$ is presented color coded in a theta – equivalent latitude coordinate system. We also added a describing text to the manuscript.

"Air masses processed in the polar vortex can also be transported to mid-latitudes. Figure 9 shows dNOy in coordinates of equivalent latitude and theta (potential temperature). Equivalent latitude takes advantage of the adiabatically quasi-conserved nature of potential vorticity. It therefore removes the variability in trace gas distributions that originates from reversible deviations from the climatological mean due to Rossby and smaller scale waves (see e.g. Hegglin et al., 2006).The early-winter period shows a relatively undisturbed distribution of reactive nitrogen, the dNOy values are close to zero.

The mid-winter period is mostly characterized by positive dNOy values, particularly above 340 K and polewards of 50° equivalent latitude. The late winter period shows a nitrified region at the same location, but with weaker nitrification than in phase II. A denitrified region is located above, predominantly at potential temperature over 380 K and equivalent latitudes over 50°. However, weak denitrification with losses up to 1 ppb is also observed throughout the whole latitude range above 360 K, even outside the vortex. Similarly, at lower isentropes slightly positive values of dNOy at lower equivalent latitudes are consistent with export of former vortex air to lower latitudes (Hoor et al., 2004, Krause et al., 2018).These findings indicate transport and mixing of vortex processed air masses to the mid-latitude lowermost stratosphere in late winter and early spring."

The following sentence was added to the abstract to refer to these results:

"Further, indications of transport and mixing of these processed air masses outside the vortex have been found, contributing to the chemical budget of the winter lower most stratosphere."

[Figure]

3) Typo:

Line 602 should read, "winter 2002/2003".

**Answer:** Done.

Answers to Comment on acp-2021-707
Anonymous Referee #2
16.12.2021

The referee's original comments are in *italics*. Our responses are written in plain black font. Changes to the manuscript text are shown in red.

We thank the referee for comments and suggestions that help to improve our manuscript.

*General comments*

*The manuscript is generally well written in a sentence-by-sentence sense, however, the text is sometimes too vague and leaves the reader guessing what the authors mean. With a few tweaks, especially in the abstract, I think the paper could be easily improved in a form that will be also appreciated by a larger group of atmospheric scientists that are not necessarily experts in reactive nitrogen in polar regions. The sentences are short and clear, however, sometimes it's hard to understand how they are connected to each other.*

**Answer:** I have tried to implement the criticisms and suggestions (see below) as best as possible to increase the readability of the text. This is especially true for the abstract.

**Specific comments**

*1) In the abstract, the authors talk about redistribution of NOy without specifying that they are talking about the vertical redistribution of NOy within the polar vortex. When tracer-tracer correlation is mentioned the author can make it clear that they are talking about N2O-NOy and N2O-O3 correlations. They talk about nitrification and de-nitrification or excess NOy and missing NOy without clearly defining with respect to what.*

**Answer:** I reformulated this section in the abstract:

"The vertical redistribution of total reactive nitrogen was evaluated by using tracer–tracer correlations ($NO_y$-$N_2O$ and $NO_y$-$0_3$). The trace gases are well correlated as long as the $NO_y$ distribution is controlled by its gas-phase production from $N_2O$. Deviations of the observed $NO_y$ from this correlation indicate the influence of heterogeneous processes. In early winter no such deviations have been observed. In January, however, air masses with extensive nitrification were encountered at altitudes between 12 and 15 km."

*2) The findings are quite clear and well presented in lines 453-456. They could be briefly summarized in the abstract as well.*

**Answer:** Yes. I have changed as suggested – see above.

*3) Line 20 "During winter 2015/2016 the Arctic stratosphere was characterized by extraordinarily low temperatures in connection with the occurrence of extensive*

*polarstratospheric clouds" mention that this is connected with a very strong polar vortex*

**Answer:** I have rephrased the sentence:

"During winter 2015/2016 the Arctic stratosphere was characterized by extraordinarily low temperatures in connection with a very strong polar vortex and with the occurrence of extensive polar stratospheric clouds."

*4) Line 26 "The redistribution of total reactive nitrogen was evaluated by using tracer– tracer correlations." Add how the correlation between N2O and NOy allows establishing ifthe airmass ia in equilibrium – denitrified or nitrified.*

**Answer:** This point was clarified, see above.

*5) Line 31: "These observations support the assumption of sedimentation and subsequent evaporation of nitric acid containing particles leading to redistribution of total reactive nitrogen" add "at lower altitudes" here*

**Answer:** I have rephrased the sentence:

"… of total reactive nitrogen at lower altitudes."

*6) Line 32: "Between end of February and mid of March also de–nitrified air masses havebeen observed in Using tracer–tracer correlations, missing total reactive nitrogen was estimated to amount up to 6 ppb. Using tracer–tracer correlations, missing total reactivenitrogen was estimated to amount up to 6 ppb. This indicates the downward transport of air masses that have been denitrified during the earlier winter phase." Move "Using tracer–tracer correlations, missing total reactive nitrogen was estimated to amount up to6 ppb" at the end of the sentence as this refers to denitrification+ high potential temperatures*

O.k. I moved this statement at the end of this section.

"… This indicates the downward transport of air masses that have been denitrified during the earlier winter phase. Using tracer–tracer correlations, missing total reactive nitrogen was estimated to amount up to 6 ppb."

*7) Line 49: the sentence "Depending on temperature, 50 composition and physical state, different types of polar stratospheric clouds can be distinguished: liquid supercooled droplets, binary or ternary solutions (SBS, STS), nitric acid hydrates (NAD, NAT) and water ice particles (e.g. Fahey et al., 2001; Hoyle et al., 2013; Khosrawi et al., 2017; Tritscher et al., 2021)." This sentence seems unnecessary/not relevant.*

**Answer:** Although, this statement has no direct implication on the following discussion, it illustrates the complexity of the heterogeneous processes in the winter polar stratosphere. Therefore, I think this might fit to the introduction.

*8) Line 53: "It does not only prepare the surface for heterogeneous reactions, it alsoremoves …" Unclear maybe use "supply" instead of "prepare"?*

**Answer:** o.k. I changed the sentence:

"… it does not only supply …"

*9) Line 55: "Heterogeneous reactions also enable the de–noxification of the stratosphere, the conversion of NOx to nitric acid" confusing. Maybe replace the comma with "by"?*

**Answer:** Done.

"... of the stratosphere by the conversion of NO$_x$..."

*10) Line 58: "The removal of nitrogen compounds from the stratosphere allows continuing ozone destruction that increases with increasing illumination of the polar vortex" the use of "increasing illumination" is not very clear maybe add "at the end of the polar winter"*

**Answer:** I agree:

"... at the end of the polar winter."

*11) Line 59: "PSCs" acronym not defined*

**Answer:** I changed the sentence.

"Polar Stratospheric Clouds (PSC)"

*12) Line 67: "UTLS" acronym not defined*

**Answer:** I changed the sentence.

"Upper Troposphere - Lower Stratosphere (UTLS)."

*13) Line 110: "So, the questions could be addressed:" change into "So, the followingquestions could be addressed:"*

**Answer:** I changed the sentence.

"So, the following questions could be addressed:"

*14) Lines 110-114: add a table in to help the reader following the timeline of*

*thecampaign*

**Answer:** I have added a small table as suggested.

| Phase I | Phase II | Phase III |
|---|---|---|
| Early winter | Mid-winter | Late winter |
| 8.12.-21.12.2015 | 12.1- 2.2. 2016 | 26.2.-18.3.2016 |

*15) Lines 179-181: add which reagent ion is used*

**Answer:** I have added:

"… using SF₅⁻ as a reagent ion."

*16) Line 184 remove extra parethesys before "Friedl"*

**Answer:** Done.

*17) Line 185 the parenthesis should be moved from before "Joahnsson" to after "et al."i.e,"discussed by Johansson et al. (2018)"*

Done.

*18) Line 214: "because their lifetime is long compared to transport time" vague sentence.Please add ranges for lifetime and transport time.*

**Answer**: I agree, this statement might be not very precise. It refers to the publication of Keim et al.: "Two species that can be considered tracers in the lower stratosphere, i.e., lifetimes of the order of decades are, nitrous oxide N2O and reactive nitrogen NOy, defined as…"

The atmospheric lifetime of N2O is more than 100 years. The exact number depends so it seems, I am not expert in this field, also on the method how this age is determined.

The various NOy components are converted into each other by photochemistry. These processes do not per se limit the lifetime of the family of reactive nitrogen compounds in the stratosphere. In general, in addition to heterogeneous processes such as PSC formation followed by sedimentation, the lifetime of NOy in the stratosphere is limited by exchange with the troposphere. Here, aerosol formation, rain-out, wash-out, or dry deposition remove NOy components from the atmosphere. Therefore, the lifetime of NOy in the lower stratosphere is determined by its exchange with the troposphere.

This links the stratospheric residence time of NOy to the transport time scales. An indication of this time scale could be the mean air age of the probed air mass. This age can be understood as the time since the last contact of the respective air mass with the troposphere (e.g., Ploeger et al., 2015). During POLSTRACC, the air mass age of the sampled air masses ranged from about 1 to 5 years (see line 294 (first draft) and Krause et al., 2018).  However, a detailed discussion is beyond the scope of this manuscript.

In my view, the important fact for the discussion in this manuscript is that NOy is formed in the stratosphere from N2O and this correlation holds as long as there are no heterogeneous processes (at least for N2O levels larger than 100 ppb). Deviations from this correlation are indications for nitrification and denitrification.

I have added these sentences to this section.

"… The relation between total reactive nitrogen and nitrous oxide can be used in this sense because their lifetime is long compared to transport time–scales (Keim et al., 1997). The lifetime of nitrous oxide is more than 100 years (Prather et al., 2015). The mean age of an air parcel can be understood as average time since

the last contact with the troposphere (Ploeger et al, 2015). During POLSTRACC the mean age of the probed air masses ranged between about 1 and 5 years (Krause et al., 2018)."

The sentence at line 239/240 was reformulated to clarify this point.

"As long as there are no additional processes, sources or sinks, in the lower stratosphere affecting the $NO_y$ concentration, observed $NO_y$ should be very close to $NO_y^*$ (within the uncertainty range of observations)."

*19) Line 221: a schematic figure of N2O vs NOy could be added to explain this.*

**Answer:** The statement in line 221 refers to the above-mentioned publication by Murphy et al. While a figure could be used for illustration, it would be a greater effort to address this point adequately. Since this is not a key message of the manuscript, an illustration would shift the focus of the manuscript. Especially since there are no other schematic illustrations in this manuscript. I will add the citation referring to Murphy et al. again at the end of the sentence to make it clear that it refers to the publication mentioned above.

In a sense, Figure 6 illustrates this relationship.

*20) Figure 1 is used to support the sentence at line 253 "As expected for undisturbed conditions, NOy and N2O are anticorrelated". For this reason, NOy and N2O should be in the same panel. Or plotted elsewhere as a scatterplot. Or at least add a vertical grid.*

**Answer:** I agree, it's difficult to see the anti-correlation in Figure 1. The anti-correlation is shown in Figure 6a. Therefore, I have moved the sentence further back in the text to the position where Figure 6 is mentioned for the first time.

"As expected for undisturbed conditions, $NO_y$ and $N_2O$ are anticorrelated (Figure 6a)."

*21) Line 235 it looks like NOy\* was determined from the least-square fit in figure 6a but in the text, it's not clear that this is the case. Add in the text (either here or at line 269) how this is used for the analysis.*

**Answer:** I have reformulated this part of the manuscript. The regression line in Figure 6 a is the result of a least squares fit. To calculate NOy\* and the deviations of the observed NOy from this value, I could have worked with the coefficients of this fit. However, I have transformed this relationship into the formulation given in equation 3 because I assume that this approach better highlights the underlying processes.

"Figure 6a shows total reactive nitrogen plotted versus $N_2O$ for the flight on 21 December. Also included in this figure is the regression line resulting from a linear least squares fit ($R^2$=0.87). The range of its uncertainty is indicated by dashed lines. As expected for undisturbed conditions, $NO_y$ and $N_2O$ are anticorrelated. To exclude tropospheric values that would affect the correlation, only values obtained in the stratosphere have been used for this analysis. In 2016 the tropospheric $N_2O$ concentration amounted about 329 ppb (Combined Nitrous

The slope of the regression line, corresponding to the factor f given in Eq. (3), is about 0.064. This value agrees reasonably well with earlier observations performed with these instruments. In late summer 2012 the HALO mission TACTS (Transport and composition in the UT/LMS) (Müller et al., 2016) was performed at northern mid latitudes. Nitrification and denitrification could be excluded for this time of the year and region. A linear least squares fit between NO$_y$ and N$_2$O for stratospheric values (N$_2$O < 320 ppb) obtained during the TACTS mission gave a slope of about 0.067. The derived slope is also comparable to findings during earlier observations in the winter Arctic region that were not affected by nitrification or denitrification. During the AASE missions in winter 1989 and 1991/1992 respectively, slopes between 0.064 and 0.078 have been observed (Fahey et al., 1990a; Fahey et al., 1990b; Weinheimer et al., 1993).

The equation describing the regression can be rewritten to take the form of equation (3). In this formulation, the following calculations of NOy* were performed. The slope obtained during the midlatitude mission TACTS was chosen as conversion efficiency f."

*22) Line 268: why a value of 320 ppb was chosen? Please add to the text.*

**Answer:** To clarify this I reformulated the text, see also above #21:

"To exclude tropospheric values that would affect the correlation, only values obtained in the stratosphere have been used for this analysis. In 2016 the tropospheric N$_2$O concentration amounted about 329 ppb (Combined Nitrous Oxide data from the NOAA Global Monitoring Laboratory). Therefore, the analysis was performed only for N$_2$O values smaller than 320 ppb."

*23) Line 269: remind the reader that this slope is the same as the "f" in equation (3) andmore in general how each term of eq3 is treated to get NOy* from the slope in Fig 6s.*

**Answer:** I clarified this point by reformulating this part of the manuscript. See above #21.

24) Line 270-276 add ranges/uncertainties to the slopes

**Answer:** This point was also addressed by referee 1. To meet these points, I added the uncertainty range arising from the linear least squares fit to Figure 1 and 6a and 7a.

See also answer to comment #21.

25) Line 278: is the value 0.067 (mid-latitude) chosen as a reference from PGS-5?

Pleaseclarify.

**Answer:** "… obtained during the TACTS mission."

*26) Line 287: "The uncertainty in the estimation of NOy\* resulting from the uncertainty of the tropospheric NOy contribution is highest directly at the tropopause and decreases with decreasing N2O concentration and increasing stratospheric character of the air mass" not obvious why this is the case. Please add an explanation in the text*

**Answer:** To clarify this point I changed the text:

"The uncertainty in the estimation of $NO_y$\* resulting from the uncertainty of the tropospheric $NO_y$ contribution is highest directly at the tropopause where the relative contribution of tropospheric NOy to NOy\* (Eq. (3)) is largest. With decreasing $N_2O$ concentration and increasing stratospheric character of the air mass NOy arising from the photooxidation of N2O increases."

*27) Line 319: "… more than 85 % of the total flight time in the lower stratosphere with PV values of more than 2 PVU" PV is not defined. Also please explain briefly in the text what it means to have a PV >2 PVU*

**Answer:** I changed the sentences:

"… with more than 85 % of the total flight time in the lower stratosphere with potential vorticity values of more than 2 PVU. The height of the dynamical tropopause is commonly attributed to the level where the potential vorticity equals this value."

*28) Line 322 add the year of TACTS*

**Answer:** I changed the sentence and added:

"… in 2012"

*29) Line 325 "Significantly higher NOy concentrations" add a value here, e.g., "up to …"*

**Answer:** I changed the sentence and added:

"…. with values up to about 10 ppb"

*30) Line 325 "than during the flight in December" add max value here*

**Answer:** I changed the sentence and added:

"… with maximum values up to 3.4 ppb"

*31) Line 334 "Values changed from around 0.004 to values up to about 0.01." Unclear ifits' referring to dNOy or to the ratios from the sentence before*

**Answer:** I have rewritten the sentence.

"Values of the $NO_y/O_3$ changed from around 0.004 to values up to about 0.01"

*32) Line 381 "As an example, the flight on 26 February (Figure 4 and 6d) may serve"change into "the flight on 26 February (Figure 4 and 6d) may serve as an example"*

**Answer:** I have rewritten the sentence:

"The flight on 26 February (Figure 4 and 6d) may serve as an example"

*33) Line 398: "Down to about 260 ppb N2O, observed NOy and calculated NOy\* agreedwithin a reasonable uncertainty range." Add uncertainty range in parenthesis*

**Answer:** I have add the following sentence:

"On average, the difference is about 0.08 ppb with a standard deviation of about 0.48 ppb."

*34) Figure 7, lower left panel: add NOy line; left panels: add a horizontal line atzero*

**Answer:** I have changed the Figure as suggested.

35) Line 549: the equation should be numbered (5) not (3)

**Answer:** *Done.*